# CoBiSyn: A Bidirectional Search Framework for Chemical Synthesis Planning

## Abstract

Artificial Intelligence is increasingly advancing scientific discovery, with chemistry being a key application domain. Synthesis planning, which aims to identify feasible reaction pathways connecting target molecules to available starting materials, is a fundamental task in organic synthesis and drug discovery. Prior work typically relies on backward search, iteratively applying single-step retrosynthesis models, which neglects information from the starting materials and often leads to inefficient exploration and redundant reactions. In this paper, we propose CoBiSyn (Coordinated Bidirectional Synthesis Planning), a framework that alternates between "backward decomposition" and "forward construction", while coordinating these two directions through shared frontier information. To support this process, we introduce a conditional embedding projection mechanism and a learned asymmetric synthetic distance, which together provide local and global cost estimates to steer the search. The experiments on multiple benchmark datasets demonstrate that CoBiSyn significantly improves the efficiency and quality for synthesis planning, compared to existing approaches.

## 1 Introduction

Recent advances in Artificial Intelligence have increasingly transformed scientific discovery, offering new paradigms for tackling complex problems in chemistry, biology, and materials science (Jumper et al., 2021; Merchant et al., 2023; Lu et al., 2024; Ding et al., 2025). Within this context, **synthesis planning**—the task of designing a feasible synthetic route for a given target molecule—represents a central task in fields such as organic synthesis and drug discovery (Blakemore et al., 2018). Since most molecules in practice cannot be obtained in a *single step*, the problem inherently involves *multi-step* planning. With the continuous emergence of novel molecular structures, traditional empirical knowledge provides limited guidance, and the exponentially expanding search space poses a significant challenge even for expert chemists (Zhong et al., 2024). A striking example is the "pupukeanane derivatives" (an important marine natural products with unique antimalarial properties): their intricate tricyclic scaffold and multiple stereocenters long defied manual retrosynthetic analysis by experienced chemists, yet plausible multi-step routes have only recently been proposed by computer-aided synthesis planning systems (Hardy et al., 2022). Such cases highlight the potential of AI to address targets beyond human intuition. In response, recent research has increasingly turned to machine learning methods for synthesis planning.

First, it is important to clarify the intended role of AI based synthesis planning tools. They are not meant to replace expert chemists, but rather to serve as assistants that generate more structured and optimized candidate hypotheses for chemists to review, refine, and validate. In practice, the model outputs should be regarded as starting points rather than fully executable solutions. As the rapid developing of AI techniques, such human-in-the-loop paradigms are widely adopted in scientific domains and have been demonstrated to improve efficiency and solution quality (M. Bran et al., 2024; Sundin et al., 2022; Watson et al., 2023) Currently, the predominant approach for synthesis planning follows a retrosynthetic reasoning paradigm, where candidate precursors are recursively predicted for the target molecule until all starting materials are drawn from the available building blocks (Segler et al., 2018; Kishimoto et al., 2019; Chen et al., 2020; Xie et al., 2022).

Although the predictive capability of single-step retrosynthesis models has been steadily improved (Chen & Jung, 2021; Zhong et al., 2023; Liu et al., 2023; Han et al., 2024), the overall

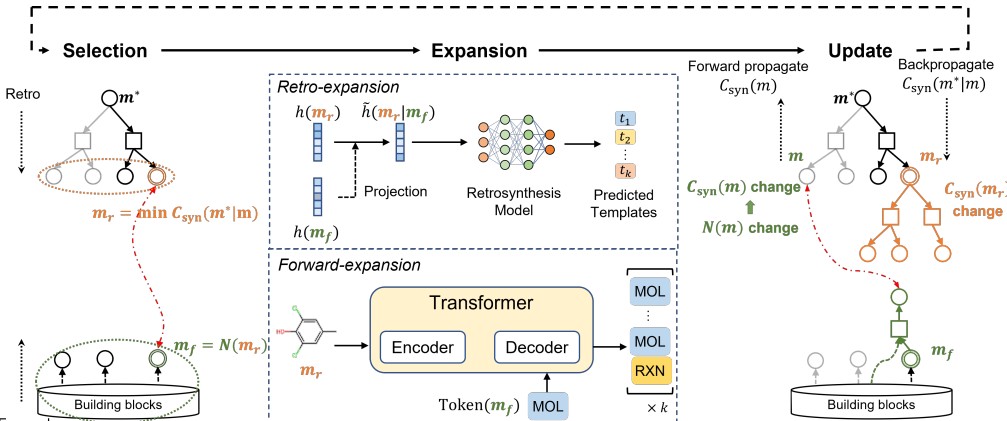

Figure 1: The overview of CoBiSyn. Circular nodes represent molecules, while square nodes denote reactions. For each internal molecule $m$, CoBiSyn maintains both intrinsic and conditional costs (denoted as $\mathbf{C}_{\mathrm{syn}}(m)$ and $\mathbf{C}_{\mathrm{syn}}(m^*|m)$, which are formally defined in Sec. 1.1). The search iterates through three phases: **(1) *Selection*:** pick boundary molecules minimizing the conditional cost of $m^*$ from both frontiers (the sets of nodes inside the dotted circles), denoting $m_r$ from the backward frontier and $m_f = N(m_r)$ as its downstream counterpart along the optimal pathway in the forward direction; **(2) *Expansion*:** apply the single-step inference model to expand the selected molecule, guided by its paired node from the opposite direction; **(3) *Update*:** propagate and revise both $\mathbf{C}_{\mathrm{syn}}(m)$ and $\mathbf{C}_{\mathrm{syn}}(m^*|m)$ to reflect the new search states.

search frameworks for **multi-step synthesis planning** have seen relatively little advancement. The widely used algorithms such as MCTS (Segler et al., 2018) and Retro* (Chen et al., 2020) still adopt a "unidirectional" reasoning process that starts solely from the target product, making it difficult to leverage information from the building block library or to obtain global guidance from the overall synthesis pathway. Consequently, the resulting routes are often unnecessarily long and may deviate from the optimal solutions. Intuitively, **reasoning about synthesis pathways resembles proving a mathematical theorem**: starting from given conditions (the starting materials), one applies a series of deductive steps (reactions) to arrive at the theorem to be proved (the target product). In practice, it is very natural to reason from both ends simultaneously—decomposing the theorem into simpler lemmas while also building complex structures from known results—until the two chains of reasoning converge. Motivated by this analogy, we ask:

*Can synthesis planning similarly benefit from a coordinated bidirectional strategy—one that explores the product and reactant sides in parallel, with each direction guided by the frontier states of the other to enable more efficient search?*

**Our main contributions.** Inspired by the above question, we aim to design a suitable bidirectional searching strategy. Nevertheless, this is goal is challenging to achieve mainly due to two reasons: first, it is unclear how to effectively integrate the forward and backward searches, as conducting them in isolation or without coordination may hinder convergence; second, most existing single-step retrosynthesis models are tailored for unidirectional expansion, making it difficult to incorporate supervisory signals from bidirectional search unless seeking for significant architectural changes.

To tackle these obstacles, we propose **CoBiSyn** (***Co**ordinated **Bi**directional **Syn**thesis Planning*), an effective search framework for chemical synthesis planning (Fig.1). Our approach alternates between retro decomposition from the product side and forward construction from the reactant side, while leveraging frontier states from both directions to guide the expansion. At each step, the framework identifies a potential counterpart molecule from the opposite side that is likely to appear on the future pathway, and leverages it as guidance to steer expansion, thereby enabling effective coordination between the two directions. To support this process, we introduce a novel design for guided expansion and cost estimation, which involves a **condition-guided embedding projection mechanism** that operates without altering the structure of single-step models, together with a **dual embedding synthetic distance** model for intrinsic and conditional cost estimation to steer the search.

The experiments on multiple benchmark datasets demonstrates that CoBiSyn achieves higher efficiency and better pathway quality than existing methods, yielding routes that are on average about

**one** reaction step shorter (approximately a 25% reduction). For instance, for a test molecule whose expert-designed synthesis pathway has a length of 10, Retro* discovers a pathway of length 7, whereas CoBiSyn achieves a pathway of just 4 steps (see Appendix F for the complete pathway). Moreover, through grouping benchmark molecules by synthesis difficulty, we find that CoBiSyn is particularly effective for challenging targets in which the competing methods frequently fail.

## 1.1 PROBLEM STATEMENT

Let $\mathbb{M} = \{m_1, m_2, \dots\}$ denote the molecular space, where each $m_i$ represents a specific molecule. Further, we let $\mathbb{B} \subset \mathbb{M}$ denote the **building blocks**, i.e., a set of available or purchasable starting materials. In this work, we adopt the concept **reaction template**: such a template defines a feasible transformation based on existing domain knowledge or chemical rules, which maps a set of molecular subgraphs of reactants to a certain product in a chemical reaction. During inference, a template serves as a candidate operator: in the retrosynthetic direction they decompose the target molecule into candidate precursors, while in the forward direction they combine suitable molecules to yield potential products. We assume $\mathbb{T} = \{t_1, \dots, t_M\}$ is the given template set. Each specific **chemical reaction** contains three components, that is, it is represented by a triplet $R = (S, m_p, t)$, where $S \subset \mathbb{M}$, $m_p \in$

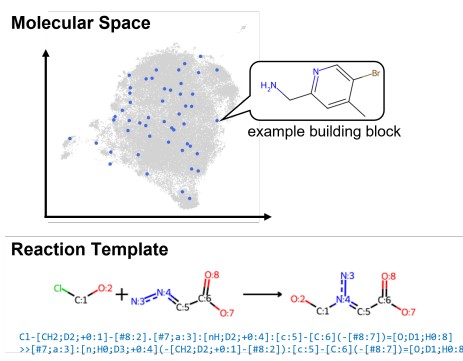

**Figure 2:** Molecular space (the grey clouds) and reaction templates define the search domain, with building blocks (the blue dot points) serving as feasible starting points. The bottom illustrates an exmaplar template of an N-alkylation reaction.

$\mathbb{M}$, and $t \in \mathbb{T}$, denoting the set of reactants, the product, and the reaction template, respectively. Fig. 2 provides an illustration for these concepts.

Given a target product $m^*$, a feasible **synthesis pathway** corresponds to a set of chemical reactions $\mathcal{P} = \{(S_i, p_i, t_i)\}$ such that they can form a directed acyclic graph with two constraints: (1) $\exists\, i$, s.t. $p_i = m^*$ (2) $\forall\, i$, if $m \in S_i$ and $m \notin \mathbb{B}$, then $\exists\, j$, s.t. $p_j = m$. The first condition ensures that the target product $m^*$ can be synthesized, while the second guarantees that all starting materials are derived from the building blocks. To assess the quality of $\mathcal{P}$, we define its cost as the sum of the costs of all starting molecules and reaction templates involved:

$$c(\mathcal{P}) = \sum_{(S, m_p, t) \in \mathcal{P}} \sum_{m \in S \cap \mathbb{B}} c(m) + \sum_{(S, m_p, t) \in \mathcal{P}} c(t), \tag{1}$$

where $c(m)$ and $c(t)$ denote the intrinsic costs of molecule $m$ and template $t$, respectively. In practice, these may correspond to molecule purchase prices and reaction expenses. Prior work typically adopts a simplified convention with $c(m) = 0$ and $c(t) = 1$, in which case $c(\mathcal{P})$ reduces to the pathway length. By contrast, CoBiSyn is a general framework that naturally accommodates richer cost formulations beyond this simplification. In our following analysis, we use $\texttt{Path}(m)$ to represent the set of all feasible synthesis pathways that can produce the molecule $m$; further, $\texttt{Path}(m|m')$ represents the subset of those pathways that contain $m'$ as an intermediate. Finally, our objective is to find a feasible synthesis pathway with minimum cost, i.e.

$$\mathcal{P}^* = \arg\min_{\mathcal{P} \in \texttt{Path}(m^*)} c(\mathcal{P}). \tag{2}$$

Upon the cost defined in (1), we introduce two specific types of costs for a molecule $m$: $\mathbf{C}_{\text{syn}}(m) = \min_{\mathcal{P} \in \texttt{Path}(m)} c(\mathcal{P})$ represents the minimum cost required to synthesize $m$, while $\mathbf{C}_{\text{syn}}(m|m') = \min_{\mathcal{P} \in \texttt{Path}(m|m')} c(\mathcal{P})$ denotes the minimum cost of synthesizing $m$ along pathways that include $m'$ as an intermediate. Here, the former captures the intrinsic synthesis difficulty of $m$, whereas the latter quantifies the conditional cost of synthesizing $m$ when an intermediate $m'$ is introduced. In particular, it is easy to see $\mathbf{C}_{\text{syn}}(m^*)$ actually is the optimal cost for synthesizing $m^*$, i.e., $\mathbf{C}_{\text{syn}}(m^*) = c(\mathcal{P}^*)$. In addition, we define $D(m, m') = \mathbf{C}_{\text{syn}}(m|m') - \mathbf{C}_{\text{syn}}(m')$ as the minimum incremental cost for synthesizing $m$ starting from $m'$, which can be regarded as an **asymmetric**

**synthetic distance** from $m'$ to $m$. In Sec. 2.4, we introduce a trainable neural network "$D_\theta$", whose output $D_\theta(m, m')$ can serve as an approximation of $D(m, m')$.

## 1.2 RELATED WORK

**Single-step retrosynthesis models** The objective of a single-step retrosynthesis model is to identify possible precursors for a given target molecule. Current approaches can be broadly categorized into three types: template-based, template-free, and semi-template methods. Template-based approaches select the most suitable transformation rule for the target molecule from a predefined set of templates (Segler et al., 2018; Dai et al., 2019; Chen & Jung, 2021; Yan et al., 2022). Template-free approaches bypass the constraints of predefined rules by modeling retrosynthesis as either a sequence generation (Liu et al., 2017; Zheng et al., 2019; Mao et al., 2021) or a graph generation problem (Sacha et al., 2021; Zhong et al., 2023). Semi-template approaches adopt a two-stage strategy: first identify the reaction center of the target molecule and decomposing it into intermediate molecules called synthons, and then complete the synthons into valid precursor molecules (Shi et al., 2020; Yan et al., 2020; Han et al., 2024).

**Synthesis planning** The mainstream approaches for retrosynthesis planning are to combine single-step retrosynthesis models with various search algorithms, iteratively identifying possible reaction transformations for the target molecule until all starting materials are sourced from the building blocks, while aiming to minimize an objective function (e.g., the number of steps or overall pathway cost). Notable methods include Proof-Number Search (Kishimoto et al., 2019), Monte Carlo Tree Search (MCTS) (Segler et al., 2018; Lin et al., 2020), and the A* algorithm (Chen et al., 2020; Xie et al., 2022). Overall, these approaches adopt a top-down strategy, starting from the target molecule and working backward to the building blocks. More recently, a few studies have explored bottom-up approaches, where synthesis routes are constructed progressively from building blocks, often in the context of synthesizable molecule design (Gao et al., 2022) or synthesizable molecule projection (Luo et al., 2024). There has also been growing interest in several variants of the synthesis planning problem, such as incorporating starting material constraints (Yu et al., 2024), accounting for uncertainties in pathway execution (Tripp et al., 2023), and integrating with active learning frameworks (Yuan et al., 2024).

**Bidirectional planners** Prior works such as DESP (Yu et al., 2024) and Tango* (Armstrong et al., 2025) also leverage bidirectional idea. Compared to these methods, our proposed method focuses on the general synthesis scenario rather than a starting-constrained setting, which dramatically increases the difficulty of search and matching when considering all available building blocks ($\sim$23M). To address this, our idea is to introduce a dual-embedding distance model for efficient cross-frontier matching and employ **explicit coordination** between forward and backward expansions to guide node exploration.

## 2 OUR PROPOSED METHOD

Before formally introducing CoBiSyn, we first present the relevant definitions that will be used throughout the subsequent sections. We follow the tradition of the previous works Chen et al. (2020); Xie et al. (2022), which models synthesis pathways as an **AND-OR graph** $G$. Specifically, $G$ consists of two alternating types of nodes: molecules represented as OR nodes and reactions represented as AND nodes. During the search, every newly added reaction $(S, m_p, t)$ corresponds to a local unit. An illustration of the graph is presented in Fig. 3. Each node is associated with a Boolean variable `solved`, indicating whether the corresponding molecule is obtainable or the reaction is executable. More specifically, we have the following formulas:

$$m.\texttt{solved} = \bigvee_{r \in \text{Child}(m)} r.\texttt{solved}$$
$$r.\texttt{solved} = \bigwedge_{m \in \text{Child}(r)} m.\texttt{solved} \tag{3}$$

where a molecule node $m$ is considered solved if at least one of its associated reaction nodes is solved, and a reaction node $r$ is solved only if all of its precursor molecule nodes are solved. For all the molecules provided by $\mathbb{B}$, their corresponding nodes are initialized with `solved = True`.

Then, we define two directions along $G$, where the **backward direction** denotes recursively decomposing the target product into simpler intermediates, and the **forward direction** indicates expanding from the starting materials toward the desired product. Please see Fig. 3 for the illustration. To enable bidirectional search, we maintain two **frontier sets**, $F_{\text{retro}}$ and $F_{\text{fwd}}$, which represent the set of molecules pending retrosynthetic decomposition and the set of molecules currently available in the forward direction, respectively. At initialization, we set $F_{\text{retro}} = \{m^*\}$ and $F_{\text{fwd}} = \mathbb{B}$, and the initial search graph $G$ contains only these two sets of nodes. Like other search-guided synthesis planning algorithms, CoBiSyn also iteratively performs three steps: *selection*, *expansion*, and *update*, until a feasible synthesis pathway is found (as shown in Fig. 1). To accommodate the bidirectional search paradigm, we introduce our specific improvements to each of these steps, which are respectively detailed in Sec. 2.1, 2.2, and 2.3. Due to the space limit, we place the full CoBiSyn algorithm to Appendix B.

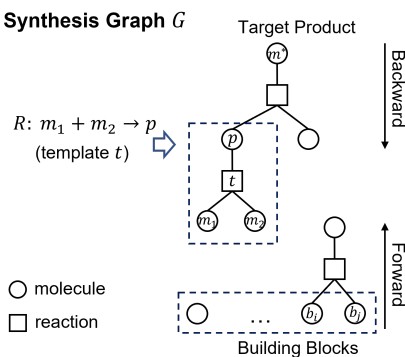

Figure 3: An AND-OR graph representation of multi-step synthesis. Circular nodes denote molecules (OR nodes) and square nodes denote templates (AND nodes). Each reaction corresponds to a local unit in graph.

### 2.1 SELECTION

To enable collaboration between the two search directions, in each iteration CoBiSyn selects one molecule from $F_{\text{retro}}$ and one from $F_{\text{fwd}}$, denoted as $m_r$ and $m_f$, respectively. One molecule can be regarded as a potential future node on the pathway of the other, serving to guide its expansion in expansion stage (details in Sec. 2.2). Since our goal is to identify synthesis pathways with minimum cost, ideally, $m_r$ and $m_f$ should both lie on the same optimal synthesis pathway. According to the definitions above, this criterion implies that $m_r$ and $m_f$ are the molecules with the minimum $\mathbf{C}_{\text{syn}}(m^*|m)$ in $f_{\text{retro}}$ and $F_{\text{fwd}}$, respectively. However, in practice, directly computing $\mathbf{C}_{\text{syn}}(m^*|m)$ is challenging, as it requires minimizing over all possible synthesis pathways (the computation on $\mathbf{C}_{\text{syn}}(m)$ also has the same issue).

Therefore, we adopt an approximation by restricting the search space to the current synthesis graph $G$ during the search. Specifically, we design a surrogate loss "$\mathbf{C}_{\text{syn}}^{\text{sur}}(m)$" to approximate $\mathbf{C}_{\text{syn}}(m)$, which can be recursively computed based on the structure of $G$: for a non-boundary node, its value is defined as the minimum cost among its subsequent reactions. Two boundary cases are considered: (1) if $m$ belongs to the building blocks, we define $\mathbf{C}_{\text{syn}}^{\text{sur}}(m) = c(m)$; (2) if $m \in F_{\text{retro}}$, we leverage the molecules in $F_{\text{fwd}}$ as intermediates and define $\mathbf{C}_{\text{syn}}^{\text{sur}}(m) = \min_{m' \in F_{\text{fwd}}} D_\theta(m, m') + \mathbf{C}_{\text{syn}}^{\text{sur}}(m')$. Here, the minimizer $m'$, denoted as $N(m)$, can be regarded as the "nearest neighbor" of $m$ in $F_{\text{fwd}}$. In summary, for any molecule node $m$ in $G$, we have:

$$\mathbf{C}_{\text{syn}}^{\text{sur}}(m) = \begin{cases} c(m) & \text{if } m \in \mathbb{B} \\ \min_{m' \in F_{\text{fwd}}} D_\theta(m, m') + \mathbf{C}_{\text{syn}}^{\text{sur}}(m') & \text{if } m \in F_{\text{retro}} \\ \min_{(S,m,t) \in G} \left[ c(t) + \sum_{m' \in S} \mathbf{C}_{\text{syn}}^{\text{sur}}(m') \right] & \text{otherwise} \end{cases} \tag{4}$$

Similarly, "$\mathbf{C}_{\text{syn}}^{\text{sur}}(m^*|m)$", a surrogate loss for $\mathbf{C}_{\text{syn}}(m^*|m)$, can be also computed in a recursive manner. We restrict the computation only to the set of molecular nodes reachable from the target molecule $m^*$ within the current search graph $G$, denoted as $V_r$, with the corresponding induced subgraph written as $G[V_r]$. The calculation is as follows:

$$\mathbf{C}_{\text{syn}}^{\text{sur}}(m^*|m) = \begin{cases} \mathbf{C}_{\text{syn}}^{\text{sur}}(m) & \text{if } m = m^* \\ \min_{\substack{(S,m_p,t) \in G[V_r] \\ \text{s.t. } m \in S}} \Big[ \mathbf{C}_{\text{syn}}^{\text{sur}}(m^*|m_p) - \mathbf{C}_{\text{syn}}^{\text{sur}}(m_p) \\ \qquad\qquad + c(t) + \sum_{m' \in S} \mathbf{C}_{\text{syn}}^{\text{sur}}(m') \Big] & \text{if } m \in V_r \setminus \{m^*\} \end{cases} \tag{5}$$

The boundary condition is specified by the target molecule itself, where $\mathbf{C}_{\mathrm{syn}}^{\mathrm{sur}}(m^*|m^*) = \mathbf{C}_{\mathrm{syn}}(m^*)$. For other molecules $m \in V_r$, the computation of $\mathbf{C}_{\mathrm{syn}}^{\mathrm{sur}}(m^*|m)$ relies on two observations: (1) any synthesis pathway in $\mathrm{Path}(m^*|m)$ must include a product molecule $m_p$ generated from $m$, enabling us to compute $\mathbf{C}_{\mathrm{syn}}^{\mathrm{sur}}(m^*|m)$ by leveraging $\mathbf{C}_{\mathrm{syn}}^{\mathrm{sur}}(m^*|m_p)$; and (2) $\mathbf{C}_{\mathrm{syn}}^{\mathrm{sur}}(m^*|m_p)$ implicitly involves $\mathbf{C}_{\mathrm{syn}}^{\mathrm{sur}}(m_p)$, but the optimal synthesis pathway of $m_p$ does not necessarily contain $m$, so we need to substitute this term with the specific cost of generating $m_p$ from $m$ (as shown in the second case of Eq. 5).

Based on the above, we can select $m_r$ from $F_{\mathrm{retro}}$ as the molecule with the minimum $\mathbf{C}_{\mathrm{syn}}^{\mathrm{sur}}(m^*|m)$ value. The corresponding $m_f$ is expected to lie downstream of $m_r$ along the optimal synthesis pathway in $\mathrm{Path}(m^*|m_r)$, which in turn guarantees that $m_r$ also belongs to the optimal pathway in $\mathrm{Path}(m_r)$. In practice, given the current search graph $G$, this $m_f$ coincides with $N(m_r)$, i.e., the molecule $m \in F_{\mathrm{fwd}}$ that minimizes $D_\theta(m_r, m) + \mathbf{C}_{\mathrm{syn}}^{\mathrm{sur}}(m)$. In summary, the selection strategy is as follows:

$$m_r = \arg \min_{m \in F_{\mathrm{retro}}} \mathbf{C}_{\mathrm{syn}}^{\mathrm{sur}}(m^*|m), \quad m_f = N(m_r) = \arg \min_{m \in F_{\mathrm{fwd}}} D_\theta(m_r, m) + \mathbf{C}_{\mathrm{syn}}^{\mathrm{sur}}(m). \tag{6}$$

To avoid redundant computation from recalculating these values at every iteration, we cache $\mathbf{C}_{\mathrm{syn}}^{\mathrm{sur}}(m)$ and $\mathbf{C}_{\mathrm{syn}}^{\mathrm{sur}}(m^*|m)$ for each molecule $m$ in $G$ and record $N(m)$ for each $m \in F_{\mathrm{retro}}$. As the search progresses, these values are dynamically updated only when necessary.

## 2.2 EXPANSION

In conventional synthesis planning approaches, the expansion step typically invokes a single-step retrosynthesis model, which performs local inference based solely on the current target molecule. Although such models can propose reasonable precursor candidates, their lack of awareness of the overall pathway structure often leads to solutions that deviate from optimal routes. In CoBiSyn, we jointly select a pair of frontier molecules $(m_r, m_f)$ from the backward and forward frontiers. When expanding one molecule, the other serves as a "conditional molecule", providing contextual guidance for single-step inference and ensuring coordinated bidirectional reasoning.

Conventional retrosynthesis model, denoted by $f_{\mathrm{single}}$, is typically formulated as a mapping from a target molecule to reaction templates, making them unsuitable for conditional-guided backward expansion. To avoid tying CoBiSyn to a specific model architecture, we do not directly modify the model itself; instead, we adjust the embedding of the target molecule by incorporating information from the conditional molecule. Conceptually, the conditional molecule serves to reweight the contributions of different substructures within the target, thereby directing the model's attention toward features most relevant to the conditional context. This adjustment can be interpreted as a projection operation in the embedding space. Based on this idea, the combined embedding is defined as

$$\tilde{h}(m_r|m_f) = A_\theta\left(h(m_f)\right) h(m_r) + B_\theta\left(h(m_f)\right), \tag{7}$$

where $h : \mathbb{M} \to \mathbb{R}^n$ is the function that maps a molecule into the embedding space (typically the encoder or the first few layers of the original model), and $A_\theta \in \mathbb{R}^{n \times n}$ and $B_\theta \in \mathbb{R}^n$ are learnable projection matrix and bias with parameter $\theta$. Thus, the **conditional single-step backward inference** model becomes

$$f_{\mathrm{retro}}(m_r|m_f) = f_{\mathrm{single}}(\tilde{h}(m_r|m_f)). \tag{8}$$

During backward expansion phase, we adopt $f_{\mathrm{retro}}(m_r|m_f)$ to predict the top-$k$ templates $\{t_i\}_{i=1}^k$, apply each templates to $m_r$ to generate the corresponding precursors $S_i$, and add the resulting nodes to the search graph $G$. For each $m \in S_i$, we initialize $\mathbf{C}_{\mathrm{syn}}^{\mathrm{sur}}(m)$ following the second case in Eq. 4.

Compared with single-step retrosynthesis model, single-step forward inference models have received relatively little attention. Inspired by Luo et al. (2024), we formulate **conditional single-step forward inference** as a sequence generation task and employ a Transformer $f_{\mathrm{fwd}}(m_f|m_r)$ with an encoder-decoder architecture to autoregressively generate candidate sequences. Specifically, a chemical reaction $(S = \{m_i\}_{i=1}^n, m_p, t)$ is represented as a sequence $[m_1, \ldots, m_n, t, m_p]$. During inference, the encoder takes $[m_r]$ as the source input, while the decoder is initialized with $[m_f]$ as the starting token. At each decoding step, the model first predicts the next token type from the logits, and then uses separate MLP heads to predict either the molecular fingerprint or the template index, depending on the predicted type. The decoding terminates once a template token is generated. The

detailed architecture of $f_{\text{fwd}}$ is provided in Appendix C. Similarly, based on the generated sequences, we perform the corresponding reactions to obtain products $\{m_{p_i}\}_{i=1}^{k}$, and add the resulting nodes to $G$. For each newly added molecule $m_{p_i}$, we initialize $\mathbf{C}_{\text{syn}}^{\text{sur}}(m_{p_i})$ according to the third case in Eq. 4.

## 2.3 UPDATE

**Update solved** After each expansion, we update the `solved` status of $m_r$ and the newly generated products $\{m_{p_i}\}$ according to Eq. 3. Whenever a node becomes `True`, this update is propagated bottom-up along the synthesis graph.

**Update frontiers** For backward expansion, the expanded node $m_r$ is removed from the frontier and the newly generated precursor molecule nodes are added to it. In contrast, forward expansion only requires adding the newly generated product nodes to the frontier, without removing any nodes, since the frontier in this case represents the set of currently available starting materials. The update rules defined as follows:

$$F_{\text{retro}} \leftarrow (F_{\text{retro}} \setminus \{m_r\}) \bigcup_{i=1}^{k} S_i, \quad F_{\text{fwd}} \leftarrow F_{\text{fwd}} \bigcup_{i=1}^{k} \{m_{p_i}\} \tag{9}$$

**Update cost** We begin by updating $\mathbf{C}_{\text{syn}}^{\text{sur}}(m)$ in a bottom-up fashion. During backward expansion, $\mathbf{C}_{\text{syn}}^{\text{sur}}(m_r)$ is updated according to the newly added reaction unit following the third case of Eq.4. During forward expansion, since new available molecules $\{m_{p_i}\}$ are introduced, the "nearest neighbor" $N(m)$ for each $m \in F_{\text{retro}}$ may change. Accordingly, we update

$$\mathbf{C}_{\text{syn}}^{\text{sur}}(m) \leftarrow \min\left\{\mathbf{C}_{\text{syn}}^{\text{sur}}(m), \min_{m_{p_i}} D_\theta(m, m_{p_i}) + \mathbf{C}_{\text{syn}}^{\text{sur}}(m_{p_i})\right\}, \quad m \in F_{\text{retro}}. \tag{10}$$

All updates to $\mathbf{C}_{\text{syn}}^{\text{sur}}(m)$ are then propagated bottom-up along the synthesis graph according to Eq.4 until they reach the target molecule $m^*$. Finally, starting from $m^*$, we perform a top-down computation of $\mathbf{C}_{\text{syn}}^{\text{sur}}(m^*|m)$ for each $m$, following Eq. 5, until reaching all nodes in $F_{\text{retro}}$.

## 2.4 MODEL TRAINING

In this framework, three models need to be trained: $f_{\text{retro}}$, $f_{\text{fwd}}$, and $D_\theta$. Due to the space limit, we describe the training procedures for $f_{\text{retro}}$ and $D_\theta$ here, while the details for $f_{\text{fwd}}$ are provided in Appendix C.

**Conditional retro model** The training of $f_{\text{retro}}$ requires triplets of the form $\mathcal{D}'_{\text{retro}} = \{(m_{\text{target}}, m_{\text{cond}}, t)\}$. Although such a dataset can be extracted from synthesis pathways (see Appendix A), it is much smaller in scale compared to the standard retrosynthesis dataset $\mathcal{D}_{\text{retro}}$. To address this, we adopt a two-stage training strategy for $f_{\text{retro}}$. In the first stage, we perform unconditional pretraining on $\mathcal{D}_{\text{retro}}$, with the projection module disabled. In the second stage, we fine-tune the model on $\mathcal{D}'_{\text{retro}}$ to learn $A_\theta$ and $B_\theta$. The pretraining stage enables the model to acquire common reaction patterns, while the fine-tuning stage teaches it to bias toward specific reactions when additional conditional cues are available.

**Distance model** As defined earlier, $D(x, y)$ denotes the minimum incremental cost of synthesizing $x$ starting from $y$, which can be regarded as a measure of the "synthetic distance" between the two molecules. To learn $D$, we adopt a dual-embedding framework. Specifically, $x$ and $y$ are encoded by two separate neural encoders, which allow for asymmetric representation learning tailored to the distinct roles of product and precursor. The resulting embeddings are mapped into a common latent space, where the $D(x, y)$ is approximated by the Euclidean distance between their embeddings:

$$D_\theta(x, y) = ||\text{Product-Encoder(x)} - \text{Precursor-Encoder(y)}||_2. \tag{11}$$

This formulation offers two key advantages. First, by employing distinct encoders for the target and precursor molecules, the model naturally accommodates the inherent asymmetry of synthetic distance, where the cost of synthesizing $x$ from $y$ generally differs from that of synthesizing $y$ from $x$. Second, the metric structure of the embedding space makes it well-suited for large-scale nearest-neighbor retrieval, which is critical to efficiently calculate $N(m)$ and $\mathbf{C}_{\text{syn}}^{\text{sur}}(m)$ for newly added node $m$ during the search (Eq. 4).

Table 1: Summary of synthesis planning efficiency across three dataset. "$n$" denotes the maximum number of single-step inference model calls allowed during the search process. The best model for each experiment setting is bolded.

| Methods | USPTO-190 | | | Pistachio Reachable | | | Pistachio Hard | | |
|---|---|---|---|---|---|---|---|---|---|
| | Solved Rate (%) ↑ | | | Solved Rate (%) ↑ | | | Solved Rate (%) ↑ | | |
| | $n$=100 | 300 | 500 | $n$=100 | 300 | 500 | $n$=100 | 300 | 500 |
| RANDOM | 20.5 | 35.3 | 40.5 | 80.0 | 88.0 | 90.0 | 39.0 | 56.0 | 57.0 |
| MCTS | 25.8 | 32.6 | 35.3 | 66.7 | 72.7 | 74.7 | 30.0 | 38.0 | 41.0 |
| Retro* | **39.5** | 45.3 | 50.0 | 90.7 | 94.7 | 96.7 | 48.0 | 57.0 | 58.0 |
| Retro*-0 | 37.9 | 46.8 | 51.6 | 90.7 | 92.0 | 95.3 | 51.0 | 53.0 | 56.0 |
| **CoBiSyn** | 38.9 | **63.2** | **68.4** | **92.0** | **96.0** | **97.3** | **53.0** | **63.0** | **69.0** |
| SimpleBiSyn | 38.9 | 54.2 | 64.2 | 86.0 | 92.0 | 96.0 | 51.0 | 60.0 | 62.0 |

The training of $D_\theta$ uses a combination of three losses. A regression term $\mathcal{L}_{reg}$ encourages predictions to match ground-truth distances, with a log-transform to handle the heavy-tailed distribution. A triangle inequality regularizer $\mathcal{L}_{triangle}$ preserves structural consistency among molecular pairs. Finally, a margin loss $\mathcal{L}_{margin}$ ensures that distances between distinct molecules are at least one reaction step. The three terms are combined with weighting coefficients to form the overall loss. Please refer to Appendix C for a detailed introduction of these losses.

## 3 EXPERIMENTS

### 3.1 EXPERIMENTAL SETUP

**Building Blocks** We adopt the list of purchasable molecules released by *eMolecules*[1] as the set of building blocks, which is widely used and provides a comprehensive and practically relevant collection of commercially available compounds. After canonicalizing the molecules with RDKit[2] and removing the entries that could not be parsed, a total of 23M molecules remain.

**Training Dataset** Although previous studies have adopted similar preprocessing pipelines, unfortunately none of them have released data in the format required for extracting training samples for conditional single-step inference networks (i.e., pathways with associated templates). Consequently, we reconstruct a synthesis pathway dataset from the USPTO reaction corpus (Lowe, 2017), following the processing procedure of Chen et al. (2020). The USPTO dataset contains approximately 3.8M published reaction records. After deduplication and template extraction with rdchiral (Coley et al., 2019) , we obtain a total of 1.42M valid reactions and 230267 reaction templates. We then randomly split the data set into training and validation sets in a 9:1 ratio. We further use these reactions to construct synthesis graphs and extract valid synthesis pathways. This process yields 235895 training routes and 27901 validation routes. Finally, we derive training and validation data for $f_{retro}$, $f_{fwd}$ and $D_\theta$ from these synthesis pathways (see Appendix A for details).

**Baseline** Since CoBiSyn is essentially a search framework for synthesis planning, we compare it against other search strategies, including RANDOM, MCTS, Retro*, and Retro*-0. RANDOM selects the next molecule node uniformly at random. MCTS (Segler et al., 2018) and Retro* (Chen et al., 2020) are both widely adopted synthesis planning algorithms: the former guides expansion via Monte Carlo rollouts, while the latter is a best-first search performed on AND-OR tree with neural-based heuristic. Retro*-0 is a variant of Retro* that does not rely on the pretrained value function as the heuristic. For fair comparison, all the methods employ the same MLP-based single-step retrosynthesis model (NeuralSyn) as in Chen et al. (2020) to predict the top-50 candidates.

### 3.2 RESULTS

We conduct the evaluations on three datasets:"USPTO-190" from Chen et al. (2020), and "Pistachio Hard" and "Pistachio Reachable" from Yu et al. (2024) (see Appendix D for an induction to these

---

[1]https://www.emolecules.com/
[2]https://www.rdkit.org/

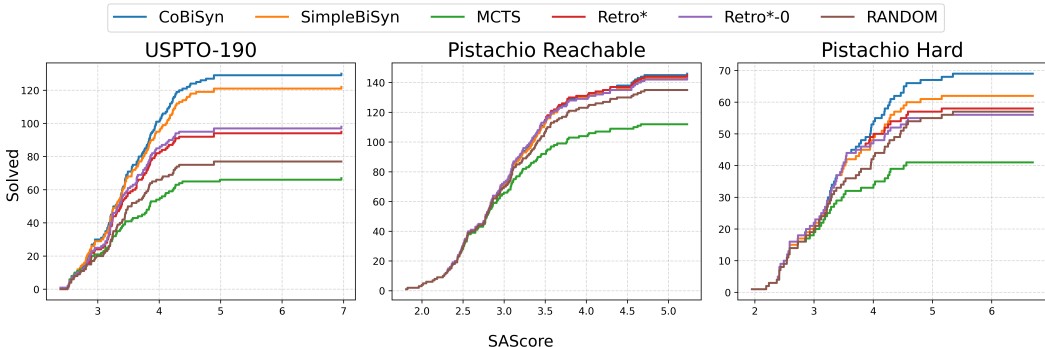

Figure 4: Cumulative solved instances on the test set ordered by SAScore. CoBiSyn shows consistent superiority, especially for molecules with SAScore > 3.

benchmarks). To assess search efficiency, we report the solved rate (the ratio of the target molecules that are successfully synthesized) under different limits on the maximum number of single-step inference model calls. The overall performance is summarized in Table 1. Compared with the baselines, CoBiSyn achieves higher solved rates across almost all thresholds on all three datasets. We also introduce an ablation variant, SimpleBiSyn, which removes the opposite-direction guidance during the expansion phase (by disabling the projection module in $f_{\text{retro}}$ and the encoder module in $f_{\text{fwd}}$). In this setting, this variant exhibits consistent performance degradation across all metrics. This clearly demonstrates the effectiveness of coordinated guidance.

To further compare those methods on handling complex molecules, we rank the test molecules by SAScore (Ertl & Schuffenhauer, 2009), a commonly used heuristic that estimates the synthetic accessibility of a molecule, and report the cumulative number of solved instances. We impose a time limit of 120 seconds per target molecule. As shown in Fig. 4, we observe that all methods perform comparably on easy cases (SAScore < 3), where synthesis routes are relatively straightforward. However, once the difficulty increases (SAScore > 3), the performance curves begin to diverge. In this regime, CoBiSyn consistently maintains the highest solved count, indicating that coordinated bidirectional search is particularly effective for tackling more challenging synthesis problems.

Furthermore, to evaluate the quality of the synthesis pathways, we report the average length of the pathways in Table 2. Across all datasets, CoBiSyn achieves substantial reductions in route length, e.g., shortens the routes by more than one step on average. We provide a concrete example in the Appendix F, where the expert-designed synthesis pathway has a length of 10, Retro* discovers a pathway of length 7, while CoBiSyn further shortens it to only 4 steps. Fig. 5 shows the individual

Table 2: The average lengths of obtained routes on common solved molecules. The best model for each experiment setting is bolded. Note that MCTS is not included here, since the number of solved molecules by MCTS is substantially lower than others (as shown in Fig. 5).

| Methods | USPTO-190 | Pistachio Reachable | Pistachio Hard |
|---------|-----------|---------------------|----------------|
| Retro* | 5.29 | 3.83 | 4.49 |
| Retro*-0 | 5.52 | 4.02 | 4.76 |
| **CoBiSyn** | **3.90** | **2.85** | **3.22** |
| SimpleBiSyn | 4.19 | 2.91 | 3.29 |

comparisons between CoBiSyn and each baseline in more detail. We observe that MCTS tends to produce relatively short routes, but the number of solved cases is limited, especially on harder datasets. In contrast, Retro* is able to solve more molecules, yet the resulting pathways are significantly longer on average. CoBiSyn achieves the best balance: it consistently shortens pathways (approximately one reaction step fewer than Retro*) while maintaining a high solved rate comparable to or better than existing methods.

Finally, we consider the sensitivity of our performance on the learned distance model $D_\theta$. In particular, we investigate how the accuracy of $D_\theta$ affects CoBiSyn 's performance. We train three distance models with identical architectures but different accuracy levels by using varying fractions of the original training data (100%, 50%, 10%). In addition, we consider a special case named "$D_\theta$-uniform", in which $D_\theta(x,y) \equiv 0$ for all input molecule pairs. This special setting means that the

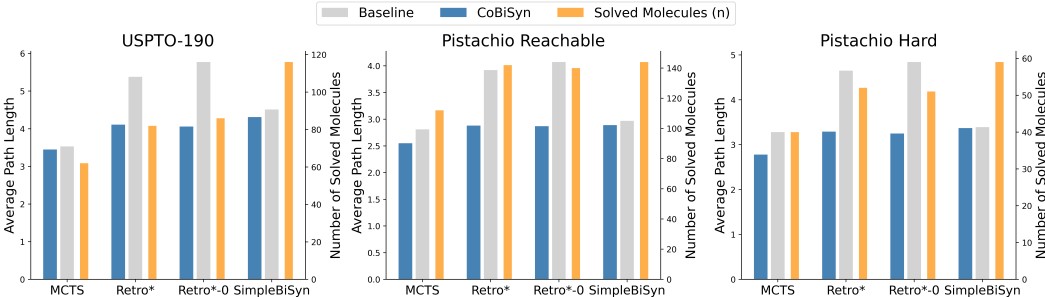

Figure 5: Comparison of CoBiSyn and baseline methods on three datasets. For each dataset, the left axis shows the average path length on molecules solved by both CoBiSyn and each baseline (blue and gray bars), while the right axis shows the number of molecules jointly solved (orange bars). CoBiSyn consistently produces shorter synthesis routes.

Table 3: Performance of CoBiSyn under distance models of varying accuracy. "$n$" denotes the maximum number of single-step inference model calls allowed during the search process. The best model for each experiment setting is bolded, and the second is underlined.

| Dist Model | USPTO-190 | | | Pistachio Reachable | | | Pistachio Hard | | |
|---|---|---|---|---|---|---|---|---|---|
| | Solved Rate (%) ↑ | | | Solved Rate (%) ↑ | | | Solved Rate (%) ↑ | | |
| | $n$=100 | 300 | 500 | $n$=100 | 300 | 500 | $n$=100 | 300 | 500 |
| 100% data | 38.9 | **63.2** | 68.4 | 92.0 | **96.0** | **97.3** | 53.0 | 63.0 | **69.0** |
| 50% data | 37.4 | 61.6 | **68.9** | 91.3 | 95.3 | **97.3** | 54.0 | 62.0 | 67.0 |
| 10% data | **42.6** | 58.4 | 66.3 | **93.3** | 95.3 | 96.0 | **56.0** | 63.0 | 66.0 |
| $D_\theta$-uniform | 32.1 | 48.9 | 55.8 | 88.0 | 92.7 | 94.7 | 49.0 | **65.0** | 67.0 |

model learns no distance information and assigns the same score to all pairs. The results of these four variants are summarized in Table 3. We observe that the model trained with 100% of the data achieves the best and most stable performance. But the performances of 50% and 10% downgrade not quite significantly. This suggest that CoBiSyn remains relatively robust even when the distance model quality is reduced. In contrast, the $D_\theta$-uniform setting leads to a clear performance degradation, particularly on USPTO-190, indicating that the learned asymmetric distance plays a non-trivial role in guiding effective cross-frontier coordination and improving search efficiency.

## 4 CONCLUSION

In this paper, we propose CoBiSyn, a framework that alternates between backward decomposition and forward construction, while coordinating the two directions through shared frontier information. Experiments on benchmark datasets demonstrate that CoBiSyn significantly improves efficiency and solution quality compared to existing approaches. It is important to note, however, that a gap remains between algorithmic performance and real-world utility, as computational evaluation metrics cannot fully capture how AI models behave in real scientific tasks. Moving forward, developing more problem-aware and scientifically grounded evaluation protocols, together with closer collaboration with chemistry experts, will be essential for translating these algorithmic advances into tangible impact in chemical synthesis.

### REPRODUCIBILITY STATEMENT

The proposed algorithm and training procedures are described in Sec. 2, with additional pre-processing procedures and implementation details provided in Appendix A-D.2. The source code and relevant data are provided in an anonymous repository at `https://github.com/anony-research/CoBiSyn-ICLR2026`.

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

# A    DATA PREPROCESSING

## A.1    REACTION FILTER

We adopt the USPTO dataset provided by `rdchiral`, which contains approximately 1.8M reactions after template extraction from the original corpus. Based on this dataset, we perform additional cleaning using the following rules:

1. Remove reactions whose product molecules contain fewer than three heavy atoms;

2. Remove reactant molecules that have no atom mappings appearing in the product;

3. Remove reactions that cannot be correctly reproduced by applying the extracted template using `RDKit` and `rdchiral`;

4. Remove reactions with multiple products.

## A.2    EXTRACT DATA FROM SYNTHESIS PATHWAYS

Following the processing procedure of Chen et al. (2020), we first extract synthesis pathway data from the reaction corpus. Based on these pathways, we further construct the training datasets required for $f_{\text{retro}}$, $f_{\text{fwd}}$, and $D_\theta$.

For training $f_{\text{retro}}$, we require samples of the form $(m, m', t)$, where $m$ is the molecule to be expanded, $t$ is a reaction template applicable to $m$, and $m'$ is the conditional molecule. Specifically, for each reaction unit $(S, m_p, t)$ in a synthesis pathway, we sample $m'$ from the set of downstream descendants of $m_p$ in the pathway, yielding training samples $(m_p, m', t)$.

Similarly, training $f_{\text{fwd}}$ requires samples of the form $(m, S, m', t)$, where $m$ is the molecule to be expanded, $S$ is the set of co-reactants, $t$ is the applicable reaction template, and $m'$ is a future molecule that provides the guiding signal. For each reaction unit $(S, m_p, t)$, we sample $m'$ from the predecessors of $m_p$, and for each $m \in S$, construct a training samples $(m, S\backslash\{m\}, m', t)$.

To train $D_\theta$, we extract molecule pairs with ground-truth distance values from constructed synthesis pathways. Given a synthesis pathway $G$ for a target molecule, we first compute $\mathbf{C}_{\text{syn}}$ for each intermediate, which indices the minimum number of reaction steps required to reach building blocks. Then for every pair of molecules $(m_x, m_y)$ where $m_y$ is reachable from $m_x$ along the pathway, we record their synthetic distance as the difference between their costs. To ensure consistency, if some pairs appear in multiple routes, we keep the minimal distance observed. This procedure produces a set of molecule pairs annotated with synthetic distances, which serve as supervision for training the $D_\theta$.

# B  FULL ALGORITHM OF COBISYN

---
**Algorithm 1** CoBiSyn

---
**Input:** Target product $m^*$
**Output:** Synthesis graph $G$
Initialize $G$ with $m^*$ and building blocks $\mathbb{B}$
**while** $m^*.solved = False$ **do**
    $m_r, m_f \leftarrow \text{SELECTION}(G)$ *// Eq. 6*

    */* Backward Expansion*                                                               *\**
    $\{t_i\}_{i=1}^k \leftarrow f_{\text{retro}}(m_r | m_f)$ *// Conditional single-step backward inference*
    **for** $i \leftarrow 1$ **to** $k$ **do**
        $S_i \leftarrow$ Apply template $t_i$ to $m_r$
        Add all $m \in S_i$ and $t_i$ to $G$
    Update $\mathbf{C}_{\text{syn}}^{\text{sur}}(m_r)$ and propagate bottom-up *// Eq. 4*
    Update $\mathbf{C}_{\text{syn}}^{\text{sur}}(m^*|m)$ for all $m$ from $m^*$ to $F_{\text{retro}}$ *// Eq. 5*

    */* Forward Expansion*                                                                   *\**
    $\{(S_i, t_i)\}_{i=1}^k \leftarrow f_{\text{fwd}}(m_f | m_r)$ *// Conditional single-step forward inference*
    **for** $i \leftarrow 1$ **to** $k$ **do**
        $m_{p_i} \leftarrow$ Apply template $t_i$ to $S_i$
        Add $m_{p_i}$ to $G$
    Update $\mathbf{C}_{\text{syn}}^{\text{sur}}(m)$ for all $m \in F_{\text{retro}}$ and propagate bottom-up *// Eq. 10 and 4*
    Update $\mathbf{C}_{\text{syn}}^{\text{sur}}(m^*|m)$ for all $m$ from $m^*$ to $F_{\text{retro}}$ *// Eq. 5*
**return** $G$

---

**Complexity Analysis**    Let $T_{\text{fwd}}$, $T_{\text{retro}}$ and $T_{\text{dist}}$ denote the per-step inference time of the forward and backward single-step reasoning models and distance model, respectively. Suppose that each inference phase attempts $k$ candidate reactions, and let $n$ denote the number of inference phases performed so far. We also let $d$ denote the dimension of the latent embedding space of the distance model (in our implementation, $d = 512$).

For the backward expansion of a molecule $m_r$, the procedure consists of three steps: (i) calling backward inference model, (ii) initializing $\mathbf{C}_{\text{syn}}^{\text{sur}}$ for the newly generated molecules, and (iii) updating $\mathbf{C}_{\text{syn}}^{\text{sur}}(m)$ and $\mathbf{C}_{\text{syn}}^{\text{sur}}(m^*|m)$ for all molecules in backward side. The resulting time complexity is

$$O(T_{\text{retro}} + \underbrace{k \cdot (|\mathbb{B}| + kn) \cdot T_{\text{dist}}}_{\substack{\text{initializing } \mathbf{C}_{\text{syn}}^{\text{sur}}(m) \\ \text{for new nodes}}} + \underbrace{kn}_{\substack{\text{updating } \mathbf{C}_{\text{syn}}^{\text{sur}}(m) \text{ and } \mathbf{C}_{\text{syn}}^{\text{sur}}(m^*|m) \\ \text{for backward side}}})$$

In practical synthesis planning settings, the building-block set is typically large (i.e., $|\mathbb{B}| \gg kn$), so the initialization term becomes a non-negligible component in addition to the inference time of expansion model. Our dual-embedding distance model is specifically designed intended to mitigate this bottleneck: the embeddings for the forward frontier are precomputed and cached, allowing each newly added molecule to query the distance model only once before performing a nearest-neighbor search. There are many efficient implementations of nearest-neighbor search exist, such as product quantization (Jegou et al., 2010) or locality-sensitive hashing Datar et al. (2004), whose cost we denote by $T_{\text{nn}}$ (see Appendix D.2). This design reduces the complexity to $O(k \cdot (T_{\text{dist}} + T_{\text{nn}}))$, where

the factor $k$ can be further mitigated through batch parallelization. Therefore, the time complexity of a single backward expansion is only $O(T_{\text{retro}} + T_{\text{dist}} + T_{\text{nn}} + kn)$.

For the forward expansion of a molecule $m_f$, the procedure also consists of three steps: (i) calling forward inference model, (ii) updating $\mathbf{C}_{\text{syn}}^{\text{sur}}(m)$ for molecules in $F_{\text{retro}}$, and (iii) updating $\mathbf{C}_{\text{syn}}^{\text{sur}}(m)$ and $\mathbf{C}_{\text{syn}}^{\text{sur}}(m^*|m)$ for all molecules in backward side. The first step takes $T_{\text{fwd}}$. The second step requires computing the embeddings for new nodes ($T_{\text{dist}}$) and calculating the distances to the nodes in the backward frontier ($O(knd)$). The third step takes $O(kn)$. Therefore, the overall time complexity of a single forward expansion is

$$O(T_{\text{fwd}} + \underbrace{T_{\text{dist}} + knd}_{\substack{\text{updating } \mathbf{C}_{\text{syn}}^{\text{sur}}(m) \\ \text{for } m \in F_{\text{retro}} \\ \text{(speed by batch parallelization)}}}).$$

In practice, $T_{\text{retro}}$ and $T_{\text{fwd}}$ correspond to more complex neural network inference operations and therefore dominate the runtime (accounting for more than 85% of the total execution time in our implementation).

## C  MODEL ARCHITECTURES AND TRAINING DETAILS

### C.1  SINGLE-STEP FORWARD INFERENCE MODEL

$f_{\text{fwd}}$ is implemented as a Transformer with an encoder-decoder architecture. The encoder takes the molecular adjacency matrix as input and processes it through multiple graph Transformer layers (Ying et al., 2021), applying multi-head self-attention over atoms to produce a tensor of atom-level embeddings.

The decoder generates tokens autoregressively by combining the encoder output with the current sequence. A reaction $(S = \{m_i\}_{i=1}^n, p, t)$ is represented as the sequence $[m_1, \ldots, m_n, t, p]$. Each molecule token is embedded using $\text{MLP}_{\text{fp}}$ via its Morgan fingerprint, while the reaction template token is embedded via an index lookup table. To capture the sequential order, positional encodings are added to all token embeddings. The sequence embeddings and molecular graph embeddings are jointly processed by several standard Transformer layers, where multi-head attention integrates the two sources of information to produce the hidden representation $h_i$ for predicting the next token.

The prediction of the next token proceeds in two stages. First, an MLP head applied to $h_i$ predicts the token type (molecule or reaction template), i.e.

$$type(T_{\text{next}}) \sim \text{SOFTMAX}(\text{MLP}_{\text{type}}(h_i)).$$

If it is a molecule, the corresponding Morgan fingerprint is predicted as $p_{\text{next}} = \text{SIGMOID}(\text{MLP}_{\text{fp-pred}}(h_i))$, followed by a nearest-neighbor search within the building blocks $\mathbb{B}$. If it is a reaction template, the index is predicted as

$$t_{\text{next}} \sim \text{SOFTMAX}(\text{MLP}_{\text{rxn}}(h_i)),$$

and the decoding process terminates.

The overall loss of $f_{\text{fwd}}$ is the sum of three parts: $\mathcal{L}_{\text{token}}$, $\mathcal{L}_{\text{mol}}$, and $\mathcal{L}_{\text{temp}}$. Here, $\mathcal{L}_{\text{token}}$ and $\mathcal{L}_{\text{temp}}$ are multi-class classification losses measured by cross-entropy, while $\mathcal{L}_{\text{mol}}$ is a binary cross-entropy loss between the predicted and ground-truth molecular fingerprints.

### C.2  TRAINING LOSSES FOR DISTANCE MODEL

The loss function of $D_\theta$ consists of three parts. The main part is to minimize the difference between the predictions and the ground truth values. Since the empirical distribution of distances extracted from synthesis pathways exhibits a heavy-tailed behavior, we apply a logarithmic transformation to stabilize training, i.e.

$$\mathcal{L}_{reg} = \mathbb{E}_{(x,y)} \left[ \text{MSE} \left( \log(D_\theta(x,y) + 1), \ \log(D(x,y) + 1) \right) \right]. \tag{12}$$

Second, we encourage consistency with the triangle inequality $D(x, y) \leq D(x, z) + D(y, z)$ by sampling triplets within each batch and penalizing violations through a hinge-style regularizer

$$\mathcal{L}_{\text{triangle}} = \mathbb{E}_{(x,y,z)} \left[ \text{RELU} \left( D_\theta(x, y) - D_\theta(x, z) - D_\theta(z, y) \right) \right]. \tag{13}$$

Preserving this property helps maintain the relative ordering among different molecular pairs, ensuring that predicted distances remain structurally coherent and chemically reasonable. Finally, to prevent degenerate solutions and reflect the fact that the distance between two distinct molecules should be at least one reaction step, we introduce a margin loss:

$$\mathcal{L}_{\text{margin}} = \mathbb{E}_{(x,y)} \left[ \text{RELU} \left( 1 - D_\theta(x, y) \right) \right]. \tag{14}$$

The final loss function is the sum of the three terms: $\mathcal{L} = \mathcal{L}_{reg} + \lambda_1 \mathcal{L}_{\text{triangle}} + \lambda_2 \mathcal{L}_{\text{margin}}$, where $\lambda_1$ and $\lambda_2$ are hyperparameters to balance three losses.

## D  EXPERIMENTS DETAILS

### D.1  INTRODUCTION TO BENCHMARKS

In our experiments, we adopt three benchmark datasets:

- **USPTO-190**: a set of 190 challenging molecules selected by Chen et al. (2020). To increase difficulty, they filter out easier molecules using a heuristic BFS planning algorithm, retaining only those unsolved within a fixed time limit.
- **Pistachio Reachable**: 150 target molecules extracted from the Pistachio dataset with specific starting materials Yu et al. (2024). They select routes satisfy: (1) no reactions appear in the training data, (2) reactions are unique across test routes, (3) all reactions are among the top 50 predictions of the single-step model, (4) no two targets share Tanimoto similarity $> 0.7$, (5) minimum number of routes enforced for different route lengths.
- **Pistachio Hard**: 100 target molecules extracted using the same procedure as Pistachio Reachable, except condition (2) is relaxed to require only $\geq 50\%$ of reactions to be reproducible (in-distribution), resulting in more challenging routes (Yu et al., 2024).

For evaluation, the starting material constraints of "Pistachio Reachable" and "Pistachio Hard" are removed.

### D.2  IMPLEMENTATION DETAILS

**Molecular representation**  In $f_{\text{retro}}$ and $D_\theta$, we use the Morgan fingerprint (Rogers & Hahn, 2010) of each molecule (radius 2 with 2048 bits) as the raw input. In contrast, model $f_{\text{fwd}}$ employs the molecular graph's adjacency matrix as input to the encoder, while molecular tokens in the decoder sequence are still represented by their Morgan fingerprints.

**Approximate nearest neighbor search**  In CoBiSyn, nearest neighbor search arises in two contexts: (1) During retro-expansion, when initializing $\mathbf{C}_{\text{syn}}$ for a newly added molecule $m$, Eq. 4 requires computing the distance $D_\theta(m, m') + \mathbf{C}_{\text{syn}}(m')$ to every molecule $m' \in F_{\text{fwd}}$ in order to identify the nearest neighbor; (2) During the decoding process of the forward expansion model $f_{\text{fwd}}$, at positions corresponding to molecular tokens, nearest-neighbor search is performed over $F_{\text{fwd}}$ based on the predicted Morgan fingerprints. Since $F_{\text{fwd}}$ contains at least all building blocks $\mathbb{B}$, its size is typically very large. To accelerate these operations, we adopt `FAISS` with Product Quantization (PQ) for approximate nearest neighbor search.

## E  OTHER EXPERIMENTAL RESULTS

### E.1  ABLATION STUDY

In this section, we provide additional ablation study setting, including: (i) removing the forward model; (ii) adopting noisy forward model (randomly combine building blocks); (iii) removing the

Table 4: Performance of CoBiSyn under different ablation setting. "$n$" denotes the maximum number of single-step inference model calls allowed during the search process. The best model for each experiment setting is bolded, and the second is underlined.

| Dist Model | USPTO-190 | | | Pistachio Reachable | | | Pistachio Hard | | |
|---|---|---|---|---|---|---|---|---|---|
| | Solved Rate (%) ↑ | | | Solved Rate (%) ↑ | | | Solved Rate (%) ↑ | | |
| | $n$=100 | 300 | 500 | $n$=100 | 300 | 500 | $n$=100 | 300 | 500 |
| CoBiSyn | 38.9 | **63.2** | **68.4** | **92.0** | **96.0** | **97.3** | **53.0** | 63.0 | **69.0** |
| w/o forward | **41.1** | 55.3 | 61.6 | 88.7 | 95.3 | 96.0 | 52.0 | 56.0 | 64.0 |
| noisy forward | 36.8 | 60.0 | 65.8 | 90.7 | 92.0 | **97.3** | 51.0 | 62.0 | 66.0 |
| w/o condition | 38.9 | 54.2 | 64.2 | 86.0 | 92.0 | 96.0 | 51.0 | 60.0 | 62.0 |
| $D_\theta$-uniform | 32.1 | 48.9 | 55.8 | 88.0 | 92.7 | 94.7 | 49.0 | **65.0** | 67.0 |

conditional projection mechanism; (iv) using a severely degraded distance model ($D_\theta(x, y) \equiv 0$). The results are summarized in Table 4.

Overall, the severe performance drop arises from degrading the distance model, as distance-based pair matching is the core operation of each expansion step and directly determines node selection and conditional projection. In contrast, the noisy forward model causes the mildest degradation. We attribute this to the forward-expansion mechanism in CoBiSyn, which accumulates newly generated nodes without discarding existing ones. Thus, even if the forward model proposes suboptimal candidates, previously valid nodes remain available for pairing with the backward frontier, preventing error from being amplified.

## E.2 CORRELATION STUDY

In this section, we investigate the factors influencing the computational consumption of CoBiSyn. In our implementation, each molecule is represented using a fixed 2048-dimensional Morgan fingerprint, which serves as input to both the expansion model and the distance model. As a result, the core computational complexity (after generating this representation) is largely independent of the molecular size. Instead, the runtime and memory usage are primarily determined by the synthetic difficulty of the target molecule. The molecules that require deeper or more exploratory search typically involve more expansion steps and frontier updates, which leads to increased computational cost.

To quantify these effects, we analyzed all target molecules in USPTO-190 with respect to (i) molecular size, measured by heavy atom count, and (ii) synthetic difficulty, measured by the length of the ground-truth synthesis pathway. Table 5 summarizes the Spearman correlation coefficients between CoBiSyn's computational cost and these molecular properties. The results indicate that runtime and search iterations exhibit only weak correlation with molecular size ($\rho = 0.413$ and $0.293$, respectively), but significantly stronger correlation with synthesis pathway length ($\rho = 0.664$ and $0.600$).

Table 5: Spearman correlation coefficients between CoBiSyn's computational cost and molecular properties. A positive coefficient indicates that the metric (runtime or search iterations) tends to increase as the corresponding property (molecular size or synthetic difficulty) increases. All $p$-values are $< 0.05$, indicating that each observed correlation is statistically significant.

| Spearman ($\rho$) | molecular size | synthetic difficulty |
|---|---|---|
| **runtime** | 0.413 | 0.664 |
| **search iterations** | 0.293 | 0.600 |

We note, however, that molecular size may still affect the overall computational complexity to some extent, independent of the used search algorithm itself. Specifically, larger molecules may incur additional overhead during RDKit-based molecule parsing and reaction template matching. They may contain more potential matching substructures, and thus the processing time can increase significantly. This explains why the Spearman correlation between molecular size and runtime ($\rho = 0.413$)

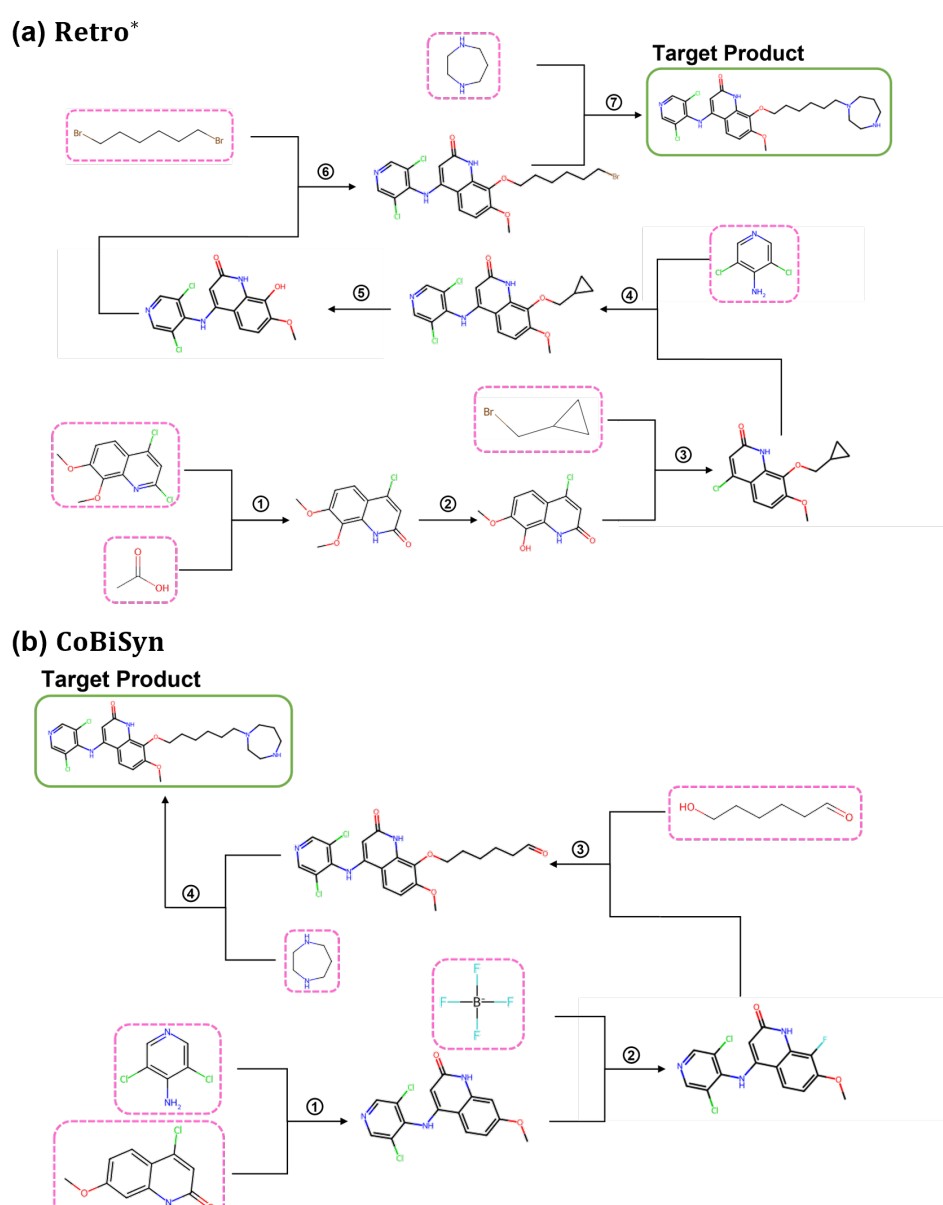

Figure 6: Illustrative examples from USPTO-190. In this case, the expert-designed route has a length of 10, Retro* yields a shorter route of length 7, while CoBiSyn further discovers an even shorter route of length 4.

is higher than that between molecular size and search iterations ($\rho = 0.293$), reflecting that the computational overhead could be caused by large molecular size rather than the CoBiSyn search algorithm itself.

## F    EXAMPLES OF SYNTHESIS PATHWAYS IDENTIFIED

Here we present two illustrative examples from USPTO-190: in the first case, CoBiSyn identifies a synthesis route that is shorter than the one found by Retro* (Fig. 6); in the second, CoBiSyn successfully identifies a valid route where Retro* fails (Fig. 7).

Figure 7: Illustrative examples from USPTO-190. In this case, the expert-designed route has a length of 8, while CoBiSyn discovers a synthesis route with length 5. Retro* fails to find a valid solution.

## G  LLM USAGE

Large Language Models (LLMs) were used in this work solely as assistive tools for polishing language of the manuscript and for debugging program errors during code development. The research ideas, experimental design, and scientific contributions were fully conceived and implemented by the authors. All authors take full responsibility for the contents of this paper.

