# OpenReview forum: "CoBiSyn: A Bidirectional Search Framework for Chemical Synthesis Planning"
_ICLR.cc/2026/Conference — Submitted to ICLR 2026_

### Official Review · Reviewer_wo5D · 2025-10-27

**Soundness:** 2
**Presentation:** 2
**Contribution:** 2
**Rating:** 2
**Confidence:** 5

**Summary:**

A bidirectional retrosynthesis framework is introduced that alternates between backward decomposition and forward construction, coordinating both directions via a shared frontier. By synchronizing these searches through common frontier information, the method yields more reliable cost estimates and improves multi-step exploration.

**Strengths:**

1. A shared frontier tightly couples backward decomposition with forward construction, improving global route consistency and multi-step exploration while avoiding single-direction myopia.

2. A condition-guided embedding projection (without modifying single-step models) plus a dual-embedding synthetic distance that separates intrinsic and condition-dependent costs yields more accurate step/route scoring, enabling better search.

3. The modules are model-agnostic and easy to integrate across reaction domains and single-step backbones.

**Weaknesses:**

Search Success Rate is a poor indicator of true performance. Your proposed forward module can game SSR, yielding scores that surpass baselines without reflecting chemical validity. This mirrors recent “synthesizable drug design” pipelines that assemble molecules bottom-up using 100+ templates: they often appear successful on paper, but many of the resulting forward reactions are chemically infeasible, revealing metric-driven illusions rather than genuine synthesis.

**Questions:**

N/A

---

> ### Author Response · Authors · 2025-11-22
> **Response to Reviewer wo5D**
>
> We really appreciate the reviewer for raising this important comment. We agree that Search Success Rate (SSR) alone does not fully reflect chemical feasibility, as reaction templates provide only a simplified abstraction of real chemical transformations, while practical synthesis often involves other experimental factors. Nevertheless, we should emphasize that both the forward and backward models in our current implementation are strictly template-based, and all proposed routes are verified for chemical validity using RDKit.
>
> Moreover, it is important to clarify the **intended role** of CoBiSyn. Synthesis planning tools are not meant to replace expert chemists, but rather to serve as **assistants** that generates more structured and optimized hypotheses for chemists to review, refine, and validate. In practice, the model output serves as a starting point rather than a final executable solution. As the rapid developing of AI techniques, such **human-in-the-loop** paradigms are widely adopted in scientific domains and have been demonstrated to improve efficiency and solution quality. For example, in the work by M. Bran et al. [1], human guidance combined with AI enabled the discovery and synthesis of a novel chromophore. Similar approaches are also commonly employed in de novo drug design [2] and protein engineering [3], where AI-generated candidates can guide experts for evaluation and decision-making.
>
> Overall, the main contribution of our work lies in a **novel algorithmic technique** for synthesis planning — by coordinating forward and backward reasoning, CoBiSyn effectively reduces search ambiguity and avoids non-promising expansion trajectories, thereby improving both the quality and efficiency of candidate route generation.
>
> Thank you again for the comments. In the revised manuscript, we have refined the "Introduction" section to more clearly clarify our contribution and its practical applicability.
>
> [1] M. Bran, A., Cox, S., Schilter, O., Baldassari, C., White, A. D., & Schwaller, P. (2024). Augmenting large language models with chemistry tools. Nature Machine Intelligence, 6(5), 525-535.
>
> [2] Sundin, I., Voronov, A., Xiao, H., Papadopoulos, K., Bjerrum, E. J., Heinonen, M., ... & Engkvist, O. (2022). Human-in-the-loop assisted de novo molecular design. Journal of Cheminformatics, 14(1), 86.
>
> [3] Watson, J. L., Juergens, D., Bennett, N. R., Trippe, B. L., Yim, J., Eisenach, H. E., ... & Baker, D. (2023). De novo design of protein structure and function with RFdiffusion. Nature, 620(7976), 1089-1100.

---

> > ### Comment · Reviewer_wo5D · 2025-11-23
> >
> > >Overall, the main contribution of our work lies in a novel algorithmic technique for synthesis planning — by coordinating forward and backward reasoning, CoBiSyn effectively reduces search ambiguity and avoids non-promising expansion trajectories, thereby improving both the quality and efficiency of candidate route generation.
> >
> > I have heard this argument many times: without a proper evaluation metric, it is impossible to reliably assess the effectiveness of an algorithm. Retrosynthetic planning is analogous to LLM text generation—each retrosynthesis step corresponds to a next-token prediction in an LLM. All leaf nodes of a predicted synthetic route are starting materials, which can be viewed as a similar role to the EOS token in LLM generation, marking the termination of the sequence. The key difference is that the search success rate only checks whether the search terminates “successfully,” whereas LLM evaluation also measures the quality of the generated sequence, for example via exact match or similar metrics.
> >
> >
> > ---
> >
> > > Nevertheless, we should emphasize that both the forward and backward models in our current implementation are strictly template-based, and all proposed routes are verified for chemical validity using RDKit.
> >
> > The template is purely match-based, and a match does not necessarily indicate correctness. From extensive manual evaluation, I found that only a very small fraction of these matches are actually correct.

---

> > > ### Author Response · Authors · 2025-11-23
> > > **Response to Reviewer wo5D**
> > >
> > > We thank the reviewer for the quick response. We agree that a template abstracts only local structural changes and does not guarantee chemical feasibility. Chemical reactions are inherently complex and sensitive to many experimental factors; even under identical conditions, different experimenters may obtain different outcomes. Nevertheless, template-based models remain the standard and widely accepted framework in computer-aided synthesis planning (e.g., AiZynthFinder, ASKCOS, SYNTHIA), as they align with the chemical intuition that local structure largely determines reactivity. Success cases such as Coley et al. [4], where AI-assisted planning enabled the synthesis of 15 drug or drug-like molecules, further illustrates that such systems can still yield chemically meaningful pathways.
> > >
> > > Within this commonly adopted abstraction, template matching provides a practical and consistent basis for generating candidate transformations and evaluating search algorithms, and our work follows this established convention. To ensure reliability, all template extraction and validation in our experiments are performed using community-standard tools such as RDKit and RDChiral. To provide an additional perspective on search performance, we also evaluate the average route length as an indicator of route quality. As shown in Table 2, CoBiSyn finds significantly shorter routes than baselines, indicating that the improvement arises from more effective navigation of the search space rather than from metric inflation.
> > >
> > > At the same time, we emphasize that CoBiSyn is designed as an assistant, not a replacement for human chemists. Given that existing reaction data cannot capture the full nuance of organic synthesis, fully autonomous retrosynthesis remains unrealistic in the near term. Instead, such systems aim to provide more structured and optimized hypotheses for chemists to review, refine, and validate. This similar perspective is also highlighted in recent literature [5].
> > >
> > > Finally, as an ICLR submission, our primary focus is on algorithmic design and methodological innovation. Improving chemical validity remains an open challenge that requires close collaboration between computer science and chemistry researchers. We appreciate the reviewer's suggestion, and we are indeed actively working with chemistry experts to bridge the gap between algorithmic advances and practical application.
> > >
> > >
> > > [4] Coley, C. W., Thomas III, D. A., Lummiss, J. A., Jaworski, J. N., Breen, C. P., Schultz, V., ... & Jensen, K. F. (2019). A robotic platform for flow synthesis of organic compounds informed by AI planning. Science, 365(6453), eaax1566.
> > >
> > > [5] Strieth-Kalthoff, F., Szymkuc, S., Molga, K., Aspuru-Guzik, A., Glorius, F., & Grzybowski, B. A. (2024). Artificial intelligence for retrosynthetic planning needs both data and expert knowledge. Journal of the American Chemical Society, 146(16), 11005-11017.

---

> ### Comment · Reviewer_wo5D · 2025-11-27
>
> Hi authors,
>
> I’m sorry to see that so many papers accepted by ICML/NeurIPS/ICLR are still using search success rate. As in my analogy between retrosynthetic planning and next-token prediction in LLMs, I don’t believe that simply “finding a route” constitutes true success; further, more rigorous evaluation is usually required. Template-based matching also introduces additional imprecision.
>
> As a researcher with many years of experience in this area, I sincerely hope the community will move away from such coarse metrics. They make it difficult to draw meaningful comparisons, and in my view, they do not adequately reflect the real quality of the methods. Evaluation, as a cornerstone of machine learning, should be as rigorous and reliable as possible.
>
> I apologize for giving a score of 2, which makes you unpleasant, but it is the reviewer's responsibility.

---

> > ### Author Response · Authors · 2025-11-27
> > **Response to Reviewer wo5D**
> >
> > We sincerely thank the reviewer for the thoughtful discussion. Just to clarify, we did not feel uncomfortable at all — we genuinely appreciate this good chance for constructive exchange.
> >
> > We completely understand your concerns. **In our view, this is indeed a common challenge faced by many AI-for-science studies**: computational evaluation metrics often fail to faithfully capture how AI models actually behave in real scientific tasks. The field needs more problem-aware and scientifically grounded evaluation protocols. Even so, we still believe **there is meaningful value in exploring algorithmic advances with these imperfections, as they often provide a practical and important starting point for pushing the field forward**.
> >
> > Going forward, we will pay closer attention to the gap between algorithmic performance and real-world utility. We will also continue strengthening our collaboration with chemistry experts to better transform these algorithmic advances into improvements that have a tangible impact in chemical synthesis. To maintain academic rigour, we just add some sentences regarding this point in the final "conclusion" section in our newly updated version (colored in blue). Thanks again!

---

### Official Review · Reviewer_FD9F · 2025-10-31

**Soundness:** 4
**Presentation:** 4
**Contribution:** 3
**Rating:** 6
**Confidence:** 4

**Summary:**

The paper proposes **CoBiSyn**, a coordinated bidirectional synthesis planning framework that jointly expands retrosynthetic and forward frontiers. Each iteration selects a pair of frontier molecules using a learned asymmetric synthetic-distance model and performs conditional single-step expansion in both directions. The method achieves better route success rates and shorter paths than strong baselines on USPTO and Pistachio-style benchmarks. The key technical novelty lies in the conditional projection mechanism that allows an existing single-step model to incorporate contextual information from the opposite frontier.

Overall, the framework is well-engineered and empirically effective. However, the conceptual contribution is **less novel than implied**. The central idea of distance-guided bidirectional search was already established by DESP and Tango, and the paper does not sufficiently acknowledge or compare against these works. The gains mainly stem from refining an existing paradigm rather than introducing a fundamentally new one.

My score is conditional on the authors addressing the above concern (Weakness).

**Strengths:**

### Strengths

1. The conditional projection mechanism is a neat and practical idea.
   It allows an existing single-step retrosynthesis model to condition on information from the opposite frontier without retraining the entire model. This design is lightweight, plug-in friendly, and demonstrates that coordination between forward and backward reasoning can be achieved through minimal architectural changes.

2. Strong empirical performance across benchmarks.
   The proposed framework achieves higher success rates and shorter routes than previous methods, under comparable rollout budgets. The improvements are consistent across multiple datasets, indicating that the approach is not just theoretically appealing but also practically effective.

**Weaknesses:**

### Weaknesses

1. Insufficient acknowledgement and comparison to prior bidirectional planners.
   The paper presents the coordinated bidirectional search as if it were a novel idea, but very similar approaches have already been proposed in DESP and Tango, which also maintain dual frontiers and employ a learned synthetic-distance model to coordinate forward and backward expansions. Therefore, the conceptual contribution is not as new as implied. The paper would be stronger if it explicitly positioned itself as a refinement of these prior bidirectional frameworks and included direct comparisons with them under comparable settings. It should also clarify which components are inherited (two-frontier maintenance, distance-guided frontier pairing) and which part is actually novel here (the conditional projection on the single-step module).

2. Heavy dependence on the learned synthetic-distance model $D_{\theta}$.
   The selection policy first chooses a backward frontier node and then uses $D_{\theta}$ to match a forward frontier node. This means that once $D_{\theta}$ misranks candidates, the two frontiers may keep expanding along mismatched branches and the search will slow down. The distance itself is trained on automatically constructed molecule pairs from the same datasets (e.g., Pistachio, which contains many short routes), so the paper should discuss how robust the method is to a less accurate distance model, whether the model needs to be retrained for different datasets, and whether an existing distance model such as the one used in DESP could be reused. As it stands, the method introduces a nontrivial prerequisite: a well-trained, task-specific distance network.

3. Forward expansion remains the weak side.
   The paper also relies on a forward one-step generator to grow routes from the starting-material side, but forward models for synthesis are much less mature than backward single-step models. Conditioning the forward generator on the opposite frontier amplifies this weakness: if the backward frontier happens to contain a suboptimal or noisy node, the forward side is guided toward that noise. The paper does not analyze performance under noisy or low-quality forward frontiers, so the overall stability of the coordinated scheme is unclear.

**Questions:**

None

---

> ### Author Response · Authors · 2025-11-22
> **Response to Reviewer FD9F**
>
> We sincerely thank the reviewer for the constructive feedback. Below, we try to address your concerns.
>
> ### Weakness 1: Insufficient acknowledgement and comparison to prior bidirectional planners
>
> Thank you for pointing out the connection to **DESP** and **Tango**. We agree that it is necessary to clarify the difference between them and our work. In our revised manuscript, we have updated the related work section to discuss these methodological similarities and to more clearly position our contribution within the context of existing bidirectional planning approaches.
>
> Overall, our problem setting, design motivations, and algorithmic choices are different from them in the following three important aspects:
>
> - *Different problem setting and complexity assumptions.*
> Both DESP and Tango assume a **starting-constrained** scenario in which the forward search frontier is initialized from a predefined starting material $s$, In such settings, the forward frontier begins as $F_f = \{s\}$, leading to a small forward frontier $F_f$ and relatively low matching cost between two directions. In contrast, our work focuses on **general** retrosynthesis planning, where the forward frontier is initialized from **all** available building blocks (23M in our experiments). This dramatically enlarges the search space and significantly increases the matching complexity.
>
> - *A distance model tailored for large-scale bidirectional matching.*
> To efficiently handle the huge number of potential pairs across two frontiers, we propose a dual-embedding distance model that maps molecules from the backward frontier $F_r$ and forward frontier $F_f$ into two separate latent spaces before computing the Euclidean distances. This architectural choice preserves the asymmetric semantics of synthetic distance, and enables efficient and highly parallelizable cross-frontier computation, which is essential for scalability and significantly differs from the distance models in DESP or Tango.
>
> - *A **coordinated** bidirectional planning framework.*
> Our proposed  **CoBiSyn** is a **coordinated** bidirectional planner, in which the two frontiers interact explicitly: the forward/backward expansion policies explicitly utilize information derived from the opposite frontier to guide node expansion. In contrast, DESP and Tango incorporate cross-frontier information only implicitly through node scoring, rather than as an explicit driver for frontier expansion. As shown in our ablation (“SimpleBiSyn” in Table 1 and 2), removing this explicit coordination mechanism leads to noticeable performance degradation, highlighting its importance for improving planning efficiency.
>
> We appreciate the reviewer’s insightful comment, which help us to better clarify our work.

---

> > ### Author Response · Authors · 2025-11-22
> > **Response to Reviewer FD9F**
> >
> > ### Weakness 2: Heavy dependence on the learned synthetic-distance model
> >
> > Thank you for raising this important comment. To assess the impact of distance models with different accuracy level, we conducted  an ablation study by training three distance models with different proportions of the original training data (100%, 50%, and 10%), yielding the models with different levels of predictive accuracy. In addition, we included a baseline named "$D_\theta$-uniform", in which $D_\theta(x,y)\equiv0$ for all input molecule pairs; this special setting means that the model provides no learned distance information and assigns identical scores to all pairs. The performance of CoBiSyn was then evaluated using each distance model, and the results are presented below.
> >
> > Table R1: Performance of CoBiSyn under distance models of varying accuracy. “n” denotes the maximum number of single-step inference model calls allowed during the search process. The best model for each experiment setting is bolded, and the second is in italics.
> > | Solved Rate (%) $\uparrow$ | USPTO (n=100) | USPTO (n=300) | USPTO (n=500) | Pis-Reach (n=100) | Pis-Reach (n=300) | Pis-Reach (n=500) |  Pis-Hard (n=100) | Pis-Hard (n=300) | Pis-Hard (n=500) |
> > |---|---|----|---|---|----|---|---|----|---|
> > | 100% Data | *38.9* | **63.2** | *68.4* | *92.0* | **96.0** | **97.3** | 53.0 | *63.0* | **69.0** |
> > | 50% Data | 37.4 | *61.6* | **68.9** | 91.3 | 95.3 | **97.3** | *54.0* | 62.0 | *67.0* |
> > | 10% Data | **42.6** | 58.4 | 66.3 | **93.3** | 95.3 | 96.0 | **56.0** | *63.0* | 66.0 |
> > |$D_\theta$-uniform| 32.1 | 48.9 | 55.8 | 88.0 | 92.7 | 94.7 | 49.0 | **65.0** | *67.0* |
> >
> > Briefly, we found that while the performance of CoBiSyn is affected by the training accuracy of the distance model, it is not overly sensitive. However, $D_\theta$-uniform leads to a noticeable drop in performance, confirming that the distance model plays a nontrivial role in guiding the planning process. The related discussion has included in the revised manuscript (section "Experiments").
> >
> > Furthermore, although the distance model proposed in DESP is conceptually similar to ours, it is not directly applicable to CoBiSyn. As we noted in our response to W1, our setting targets general synthesis planning, where the number of forward-backward node pairs in the bidirectional search (~23M) is much larger than  that in the scenarios considered by DESP. DESP’s distance model is not optimized for such large-scale pairwise evaluations, and directly applying it would require an enormous number of model calls, which would lead to inefficiencies in node scoring.
> >
> > ### Weakness 3: Forward expansion remains the weak side
> >
> > We thank the reviewer for pointing out the potential sensitivity to low-quality forward predictions. From a methodological perspective, CoBiSyn accumulates newly generated forward nodes into the frontier without removing current nodes (as illustrated in Eq. (9)). Therefore, even if the forward model proposes suboptimal candidates, previously valid nodes remain available for pairing with backward frontiers, and thus mitigate the error amplification. To validate this, we conducted an additional experiment simulating a severely degraded forward predictor: at every forward expansion step, the model output is ignored and molecules are instead randomly combined from the forward frontier. As shown in the results below, injecting noisy molecules into the forward frontier leads to only a limited performance drop. We further compare the performance with removing the forward module, which decreases more across almost all settings, particularly for USPTO-190 and Pistachio-Hard, where the target molecules are more challenging to synthesize. These results indicate that forward-side candidate expansions from starting materials are indeed beneficial for route search, as they provide more opportunities for steering the backward expansion. We have included the related comparisons in Appendix E in our revised manuscript.
> >
> > Table R2. Performance comparison between CoBiSyn and the versions using noisy forward module or removing forward module.
> > | Solved Rate (%) $\uparrow$ | USPTO (n=100) | USPTO (n=300) | USPTO (n=500) | Pis-Reach (n=100) | Pis-Reach (n=300) | Pis-Reach (n=500) |  Pis-Hard (n=100) | Pis-Hard (n=300) | Pis-Hard (n=500) |
> > |--|---|---|---|---|----|---|---|---|---|
> > |CoBiSyn| 38.9 | **63.2** | **68.4** | **92.0** | **96.0** | **97.3** | **53.0** | **63.0** | **69.0** |
> > |Noisy Forward | 36.8 | 60.0 | 65.8 | 90.7 | 92.0 | **97.3** | 51.0 | 62.0 | 66.0 |
> > |Remove Forward | **41.1** | 55.3 | 61.6 | 88.7 | 95.3 | 96.0 | 52.0 | 56.0 | 64.0 |

---

> > > ### Author Response · Authors · 2025-11-27
> > >
> > > Dear Reviewer `FD9F`,
> > >
> > > Thank you for taking the time to review our paper and provide valuable feedback. As the discussion phase is nearing its conclusion, we would like to confirm if our responses from a few days ago have effectively addressed your concerns. If you have any additional comments, we will do our best to address them.
> > >
> > > Best regards,
> > >
> > > The authors of Submission **16488**.

---

> ### Comment · Reviewer_FD9F · 2025-11-28
>
> Thank you for the detailed author response. I believe the authors have adequately addressed my concerns, and I now have a clearer understanding of the motivation and methodological contributions. The motivation appears genuinely novel, and the ablation studies and evaluations provided fall within a reasonable and sufficient scope for an ML-oriented framework.
>
> I will keep my original score, as my evaluation was already based on the assumption that the authors would successfully address the raised concerns.

---

> > ### Author Response · Authors · 2025-11-28
> > **Response to Reviewer FD9F**
> >
> > We would like to thank the reviewer again for the helpful comments and suggestions.

---

### Official Review · Reviewer_f5Z7 · 2025-11-01

**Soundness:** 3
**Presentation:** 3
**Contribution:** 2
**Rating:** 6
**Confidence:** 3

**Summary:**

This paper proposes CoBiSyn, a coordinated bidirectional search framework for multi-step chemical synthesis planning that alternates between backward decomposition and forward construction while sharing frontier information through conditional embedding projection and an asymmetric synthetic distance heuristic, achieving higher efficiency and route quality than unidirectional baselines.

**Strengths:**

* The paper introduces a novel and well-justified bidirectional search paradigm for synthesis planning.

* The framework elegantly coordinates forward and backward reasoning through conditional embeddings and learned distance metrics.

* The proposed method achieves consistent and significant improvements across multiple benchmark datasets and shows strong potential impact for AI-driven drug discovery and materials design.

**Weaknesses:**

* The paper does not include comparisons with forward-only or hybrid search baselines.

* The proposed method lacks quantitative analysis of computational cost and scalability.

* The performance of CoBiSyn heavily depends on the accuracy of the learned distance model.

* The paper provides limited discussion of the interpretability of the learned representations.

**Questions:**

* How does the framework determine when to alternate between forward and backward search steps?

* How sensitive is the performance of CoBiSyn to the selection of the asymmetric distance model?

* Can this method be extended to handle these synthesis tasks with multiple targets?

* What is the runtime and memory overhead of CoBiSyn on large molecules?

---

> ### Author Response · Authors · 2025-11-22
> **Response to Reviewer f5Z7**
>
> We sincerely thank the reviewer for the constructive feedback. Below, we try to address your concerns.
>
> ### Q1: How does the framework determine when to alternate between forward and backward search steps?
>
> In our current implementation, the search process alternates between forward and backward expansion in a fixed manner; specifically, each iteration performs one backward step followed by one forward step (as described in Alg. 1 of Appendix B). We adopted this simple alternating schedule mainly because it is easy to implement and worked consistently well across our benchmark settings. We agree that more adaptive or learned switching policies may have the chance to further improve efficiency and robustness. Thank you for this question, and we think it is an interesting direction deserved future investigation.
>
> ### Q2 & Weakness 3: How sensitive is the performance of CoBiSyn to the selection of the asymmetric distance model?
>
> Thanks for this constructive question. To assess the sensitivity of CoBiSyn to the choice and quality of the asymmetric distance model, we conducted an ablation study by training three distance models with different proportions of the original training data (100%, 50%, and 10%), yielding the models with different levels of predictive accuracy. In addition, we included a baseline named "$D_\theta$-uniform", in which $D_\theta(x,y)\equiv0$ for all input molecule pairs; this special setting means that the model provides no learned distance information and assigns identical scores to all pairs. The performance of CoBiSyn was then evaluated using each distance model, and the results are presented below.
>
> Table R1: Performance of CoBiSyn under distance models of varying accuracy. “n” denotes the maximum number of single-step inference model calls allowed during the search process. The best model for each experiment setting is bolded, and the second is in italics.
> | Solved Rate (%) $\uparrow$ | USPTO (n=100) | USPTO (n=300) | USPTO (n=500) | Pis-Reach (n=100) | Pis-Reach (n=300) | Pis-Reach (n=500) |  Pis-Hard (n=100) | Pis-Hard (n=300) | Pis-Hard (n=500) |
> |---|---|----|---|---|----|---|---|----|---|
> | 100% Data | *38.9* | **63.2** | *68.4* | *92.0* | **96.0** | **97.3** | 53.0 | *63.0* | **69.0** |
> | 50% Data | 37.4 | *61.6* | **68.9** | 91.3 | 95.3 | **97.3** | *54.0* | 62.0 | *67.0* |
> | 10% Data | **42.6** | 58.4 | 66.3 | **93.3** | 95.3 | 96.0 | **56.0** | *63.0* | 66.0 |
> |$D_\theta$-uniform| 32.1 | 48.9 | 55.8 | 88.0 | 92.7 | 94.7 | 49.0 | **65.0** | *67.0* |
>
> We observe that  the model trained with 100% of the data achieves the best and most stable performance. But the  performances of 50% and 10% downgrade not quite significantly. This suggest that CoBiSyn remains relatively robust even when the distance model quality is reduced. In contrast, the $D_\theta$-uniform setting leads to a clear performance degradation, indicating that the learned asymmetric distance plays a non-trivial role in guiding effective cross-frontier coordination and improving search efficiency. We have incorporated the corresponding discussion in the revised manuscript (section "Experiments").
>
> ### Q3: Can this method be extended to handle these synthesis tasks with multiple targets?
>
> Yes, CoBiSyn can be extended to optimize multiple targets (e.g. cost, route length, availability) during planning. For example, we can use a weighted linear combination of multiple objectives (which is a commonly adopted approach in multi-objective optimization), and then our framework can naturally accommodate such a formulation by modifying "$c(m)$" and "$c(t)$" defined in Eq. (1). Unfortunately, such information for measuring those targets are often proprietary to industry companies and not publicly available, so we cannot conduct more experiments to further validate this extension in current stage. But we still thank the reviewer for raising this question, and we are more than happy to investigate this problem as long as such multi-target datasets are available in future.

---

> > ### Author Response · Authors · 2025-11-22
> > **Response to Reviewer f5Z7**
> >
> > ### Q4: What is the runtime and memory overhead of CoBiSyn on large molecules?
> >
> > Thanks for this question. In fact, the size of a molecule is not a major factor for determining the runtime or memory consumption of CoBiSyn. In our current implementation, each molecule is represented using a fixed 2048-dimensional Morgan fingerprint as the input to both the expansion model and the distance model, so the computational complexity (after the representation step) should be independent of the input size. Actually, the runtime and memory usage are primarily influenced by the **synthetic difficulty** of the target molecule. Molecules requiring deeper or more exploratory search typically involve more expansion steps and frontier updates, which leads to increased computational cost.
> >
> > To further investigate their correlations, we analyzed all the target molecules in USPTO-190 using (i) molecular size (measured by heavy atom count) and (ii) synthetic difficulty (measured by ground-truth synthesis pathway length). The results, shown below, indicates that runtime and search iterations exhibit only weak correlation with molecular size ($\rho$ = 0.413 and 0.293, respectively), but significantly stronger correlation with synthesis pathway length ($\rho$=0.664 and 0.600).
> >
> >
> > Table R2: Spearman correlation coefficients between CoBiSyn’s computational cost and molecular properties. A positive coefficient indicates that the metric (runtime or search iterations) tends to increase as the corresponding property (molecular size or synthetic difficulty) increases. All p-values are < 0.05, indicating that each observed correlation is statistically significant.
> > | Spearman ($\rho$) | molecular size | synthetic difficulty |
> > |---|---|---|
> > | runtime | 0.413 | 0.664 |
> > | search iterations | 0.293 | 0.600 |
> >
> >
> > However, we should acknowledge that molecular size may still affect the overall computational complexity to some extent, independent of the used search algorithm itself. Specifically, larger molecules may incur additional overhead during RDKit-based molecule parsing and reaction template matching. They may contain more potential matching substructures, and thus the processing time can increase significantly. This explains why the Spearman correlation between molecular size and runtime ($\rho$=0.413) is higher than that between molecular size and search iterations ($\rho$=0.293), reflecting that the computational overhead could be caused by large molecular size rather than the  CoBiSyn search algorithm itself.
> >
> > We include the above discussion to Appendix G in the revised manuscript.
> >
> > ### Weakness 1: does not include comparisons with forward-only or hybrid search baselines
> >
> > Thanks for your suggestion. We indeed have evaluated forward-only search across our benchmarks, but it fails to find valid routes for nearly all targets. This outcome is foreseeable, as current forward single-step models are significantly less mature than backward single-step models. Nevertheless, in CoBiSyn, the forward module still plays a critical role: it provides candidate expansions from the starting materials, which are then paired to backward frontier nodes and guided via conditional projection, thereby enhancing their value and contributing to successful route discovery. We conducted the ablation experiments in which the forward generator was removed.
> >
> >
> > Table R3. Performance comparison between CoBiSyn and the version without the forward module.
> > | Solved Rate (%) $\uparrow$ | USPTO (n=100) | USPTO (n=300) | USPTO (n=500) | Pis-Reach (n=100) | Pis-Reach (n=300) | Pis-Reach (n=500) |  Pis-Hard (n=100) | Pis-Hard (n=300) | Pis-Hard (n=500) |
> > |--|---|---|---|---|----|---|---|---|---|
> > |CoBiSyn| 38.9 | **63.2** | **68.4** | **92.0** | **96.0** | **97.3** | **53.0** | **63.0** | **69.0** |
> > |Remove Forward | **41.1** | 55.3 | 61.6 | 88.7 | 95.3 | 96.0 | 52.0 | 56.0 | 64.0 |
> >
> >
> > As shown above, the performance decreases across almost all settings, particularly for USPTO and Pistachio-Hard, where the target molecules are more challenging to synthesize (only for USPTO (n=100), the performance of "Remove Forward" is slightly better than "CoBiSyn"). These results underscore the importance of the forward module in enabling effective cross-frontier coordination. The related discussion including other hybrid search comparisons has been added in Appendix G of the revised manuscript.

---

> > > ### Author Response · Authors · 2025-11-22
> > > **Response to Reviewer f5Z7**
> > >
> > > ### Weakness 2: lacks quantitative analysis of computational cost and scalability
> > >
> > > Thanks for your suggestion. We have appended the related discussion in Appendix B in our revised manuscript. Let $T_{fwd}$, $T_{retro}$ and $T_{dist}$ denote the **per-step** inference time of the forward and backward single-step reasoning models and distance model, respectively. We also let $d$ denote the dimension of the latent embedding space (in our implementation, $d=512$).
> > >
> > > A single backward expansion consists of three steps: calling the backward model, finding the pairings for the new nodes, and updating the relevant variables of nodes on the backward side of the search graph (as described in Alg. 1 of Appendix B). The first step takes $T_{retro}$. For the second step, due to our dual-embedding distance model, we only need to compute the embeddings for the new nodes once, followed by a nearest-neighbor query over the pre-cached embeddings of starting materials. We can apply existing efficient implementations for nearest-neighbor search, such as the popular product quantization or locality-sensitive hashing; in our experiments, we use product quantization and denote this cost as $T_{nn}$. Thus, the second step yields a total cost of $O(T_{dist} + T_{nn})$. The third step updates the node variables on the backward side of search graph. Assuming each expansion attempts $k$ reactions (a predefined hyper-parameter) and therefore generates $O(k)$ new nodes. Let $n$ denote the number of expansion iterations performed so far, and then the backward search graph contains $O(nk)$ nodes. Node variable updates can be implemented via one bottom-up and one top-down broadcast (as introduced in Sec. 2.3), with a cost of $O(nk)$. Overall, the time complexity of a single backward expansion is:
> > >
> > > $$
> > > O(T_{retro} + T_{dist} + T_{nn} + nk)
> > > $$
> > >
> > > Similarly, a single forward expansion consists of three steps: calling the forward model, updating the pairings for nodes in the backward frontier, and updating the relevant variables of nodes on the backward side. The first step takes $T_{fwd}$. The second step requires computing the embeddings for new nodes ($T_{dist}$) and calculating the distances to the nodes in the backward frontier ($O(nkd)$). The third step takes $O(nk)$. Therefore, the overall time complexity of a single forward expansion is:
> > >
> > > $$
> > > O(T_{fwd} + T_{dist} + nkd)
> > > $$
> > >
> > > In practice, $T_{retro}$ and $T_{fwd}$ correspond to more complex neural network inference operations and therefore dominate the runtime (accounting for more than 85% of the total execution time in our implementation). The above discussion has been included in Appendix B of revised manuscript.
> > >
> > > ### Weakness 4: limited discussion of the interpretability of the learned representations
> > >
> > > In our representation part, the key is the conditional projection idea, where the motivation is to inject directional guidance into the retrosynthesis model through a lightweight and controllable manner. Its design intuition relies on the embedding space: a retrosynthesis model should predict the templates based on the molecular embedding. If we hope to steer the expansion toward certain regions in the search space (e.g., consistent with the opposite frontier), we can achieve this by reweighting different latent features of the embedding. From a geometric perspective, such reweighting corresponds to projecting the original embedding onto a directionally biased subspace. Therefore, we introduce a learnable projection layer whose direction and offset are conditioned on the opposite-frontier signal (the intuition is briefly discussed in Sec 2.2).  Our ablation study ("SimpleBiSyn" in Table 1 and 2) further confirms the importance of such directional conditioning, as removing the projection markedly reduces the route quality and search efficiency.

---

> > > > ### Author Response · Authors · 2025-11-27
> > > >
> > > > Dear Reviewer `f5Z7`,
> > > >
> > > > Thank you for taking the time to review our paper and provide valuable feedback. As the discussion phase is nearing its conclusion, we would like to confirm if our responses from a few days ago have effectively addressed your concerns. If you have any additional comments, we will do our best to address them.
> > > >
> > > > Best regards,
> > > >
> > > > The authors of Submission **16488**.

---

### Author Response · Authors · 2025-11-22
**General Response**

We sincerely thank all the reviewers for their  valuable comments. In response, we have added new content to the revised manuscript. All the changes are marked in **blue** within the paper and summarized below:

- **Page 1, Lines 43-49**: We add a paragraph clarifying the role of AI synthesis planning tools as human-in-the-loop assistants and their practical usage paradigm (in response to reviewer `wo5D`).
- **Page 4, Lines 190-197**: We add a comparison with other bidirectional planner and clarify our contribution (for **Weakness 1** from reviewer `FD9F`).
- **Page 10, Lines 502-525**: We add an ablation study to evaluate the sensitivity of distance model in CoBiSyn (for **Q2 & Weakness 3** from reviewer `f5Z7` and **Weakness 2** from reviewer `FD9F`).
- **Page 14, Lines 736-772**: We add the complexity analysis for CoBiSyn (for **Weakness 2** from reviewer `f5Z7`).
- **Page 16, Lines 861-887**: We add additional ablation study to investigate the influence of different forward model settings (for **Weakness 1** from reviewer `f5Z7` and **Weakness 3** from reviewer `FD9F`).
- **Page 18, Lines 889-964**: We add a correlation study between computational cost and molecular properties (for **Q4** from reviewer `f5Z7`).

---

### Author Response · Authors · 2025-12-04
**A Brief Summary of Rebuttal**

Dear PCs, SACs, ACs, and Reviewers,

We sincerely appreciate the time and effort dedicated to reviewing our submission. This paper falls within the area of AI for science, focusing on **synthesis planning**, a central problem in organic synthesis and drug discovery that aims to identify feasible reaction pathways for given target molecules. First, we are pleased that several **strengths** of our work were recognized by the reviewers:

- The research motivation is novel and well-justified (Reviewer `f5Z7`, `FD9F`).

- The framework effectively coordinates forward and backward reasoning through well-engineered design (Reviewer `f5Z7`, `FD9F`, `wo5D`).

- The proposed modules are lightweight, model-agnostic, and easy to integrate across reaction domains and single-step backbones (Reviewer `FD9F`, `wo5D`).

- The method demonstrates strong and consistent empirical performance across multiple benchmarks (Reviewer `f5Z7`, `FD9F`).

- The method shows strong potential impact for AI-driven synthesis planning, drug discovery, and materials design (Reviewer `f5Z7`).


--------

### **Summary about the feedbacks during the discussion:**

1) We are pleased that **Reviewer `FD9F`** confirmed that the concerns were fully resolved (https://openreview.net/forum?id=t0Dh91HXdQ&noteId=h5ouwcFrs6).

2) **Reviewer `f5Z7`** did not provide a response before the shutdown of discussion.

3) We have a multi-round discussion with **Reviewer `wo5D`** to explain the evaluation metric and the intended role of our "CoBiSyn" in real world.

First, we understand Reviewer `wo5D`'s concern. As we explained in our response (https://openreview.net/forum?id=t0Dh91HXdQ&noteId=tSGiST9hUn), "Chemical reactions are inherently complex and sensitive to many experimental factors; even under identical conditions, different experimenters may obtain different outcomes." In other word, it is hard to find a perfect evaluation metric to faithfully capture how AI models actually behave in real scientific tasks. Even the evaluation by real chemical experiment may not be fully reliable, since it can be influenced by the lab's environment (e.g., temperature and humidity) and the experimenter's level of expertise.

Nevertheless, template-based models remain the **standard** and **widely accepted** paradigm in computer-aided synthesis planning (e.g., the popular open-source softwares "AiZynthFinder" and "ASKCOS"), as they align with the chemical intuition that local structure largely determines reactivity. Moreover, we follow the **community-standard tools** (RDKit, RDChiral) for template extraction and reaction validation to **ensure reliability and comparability**.

Finally, the purpose of CoBiSyn is to serve as an **assistant**, not a replacement for expert chemists.  In practice, we should follow a **human-in-the-loop** workflow: the model proposes structured hypotheses, and then chemists refine and validate them. For more detailed explanation, please refer to our responses https://openreview.net/forum?id=t0Dh91HXdQ&noteId=ucj5Ir5eeS, https://openreview.net/forum?id=t0Dh91HXdQ&noteId=tSGiST9hUn, and https://openreview.net/forum?id=t0Dh91HXdQ&noteId=ECQTSOi3Hx

-------

Overall, we hope the above summary could assist the newly assigned AC in evaluating our work. Also, we are deeply grateful to the reviewers, ACs, SACs and PCs for their time, insightful feedback, and dedicated efforts throughout this process.

Best regards,

The authors of Submission **16488**.

---

### Meta-Review · Area_Chair_5Qdt · 2026-01-06

**Summary:**

This paper proposes CoBiSyn, a coordinated bidirectional synthesis planning framework that alternates between backward retrosynthetic decomposition and forward construction from starting materials, with coordination achieved via a conditional embedding projection mechanism and a learned asymmetric synthetic-distance model. The method aims to improve search efficiency and route quality compared to traditional backward-only planners. Experiments on USPTO and Pistachio-style benchmarks show higher search success rates and shorter routes under fixed rollout budgets.

Reviewers acknowledged that the framework is well engineered, that the conditional projection mechanism is practical and lightweight, and that empirical results demonstrate improvements over several baselines. The authors were responsive in the rebuttal and added ablations, complexity analyses, and clarifications that addressed a number of technical questions. However, **significant concerns remain regarding  the reliability of the primary performance metric**. In particular, concerns raised by Reviewer wo5D about the inadequacy of search success rate (SSR) as an evaluation metric were not resolved, which fundamentally undermines the strength of the empirical evidence. These unresolved issues motivate the rejection decision.

**Reviewer Concerns:**

Concerns that have been addressed satisfactorily:
- In response to concerns about computational cost and scalability raised by Reviewer f5Z7: the authors added a detailed complexity analysis and empirical correlations between runtime, molecular size, and synthetic difficulty, clarifying the scalability characteristics of CoBiSyn.
- In response to concerns about sensitivity to the learned synthetic-distance model raised by Reviewers f5Z7 and FD9F: the authors conducted ablation studies using distance models trained with varying data fractions and a uniform-distance baseline, showing that performance degrades gracefully and confirming the nontrivial role of the distance model.
- In response to concerns about comparison with forward-only or hybrid variants raised by Reviewer f5Z7: the authors added ablations removing or corrupting the forward module, demonstrating that coordinated bidirectional expansion improves performance relative to these variants.
- In response to concerns about overlap with prior bidirectional planners raised by Reviewer FD9F: the authors revised the related work section and clarified differences from DESP and Tango, particularly in problem setting, scalability assumptions, and explicit cross-frontier coordination.

Concerns that have not been addressed satisfactorily:
- In response to concerns about conceptual novelty raised by Reviewer FD9F: although distinctions from prior work were clarified, the method remains a refinement of existing bidirectional planning paradigms rather than a fundamentally new approach. The gains largely stem from improved coordination and heuristics, which limits the perceived level of contribution.
- In response to concerns about evaluation validity raised by Reviewer wo5D: the reviewer argued that search success rate (SSR) is an inadequate and potentially misleading metric, as template-based matching can inflate apparent success without reflecting true chemical feasibility. While the authors acknowledged these limitations and emphasized community conventions and human-in-the-loop usage, they did not introduce stronger or alternative evaluation metrics that directly address this concern. As a result, the core objection regarding the reliability of the empirical evaluation remains unresolved.

**Reviewer Scores:**

- Reviewer f5Z7: Marginally positive (6); would likely maintain the score, remaining open to rejection despite clarified ablations and analyses.
- Reviewer FD9F: Marginally positive (6); would maintain the original score, having found the rebuttal adequate but already conditional on acceptance.
- Reviewer wo5D: Strongly negative (2); would maintain the score, as concerns about evaluation metrics and chemical validity were not resolved.

---

### Decision · Program_Chairs · 2026-01-26

Reject